# Combination of inflammatory and vascular markers in the febrile phase of dengue is associated with more severe outcomes

Nguyen Lam Vuong[1,2]\*, Phung Khanh Lam[1,2], Damien Keng Yen Ming[3], Huynh Thi Le Duyen[1], Nguyet Minh Nguyen[1], Dong Thi Hoai Tam[1], Kien Duong Thi Hue[1], Nguyen VV Chau[4], Ngoun Chanpheaktra[5], Lucy Chai See Lum[6], Ernesto Pleités[7], Cameron P Simmons[8,9], Kerstin D Rosenberger[10], Thomas Jaenisch[10,11], David Bell[12], Nathalie Acestor[13], Christine Halleux[14], Piero L Olliaro[8], Bridget A Wills[1,8], Ronald B Geskus[1,8], Sophie Yacoub[1,8]\*

[1]Oxford University Clinical Research Unit (OUCRU), Ho Chi Minh City, Viet Nam; [2]University of Medicine and Pharmacy at Ho Chi Minh City, Ho Chi Minh City, Viet Nam; [3]Department of Infectious Disease, Imperial College London, London, United Kingdom; [4]Hospital for Tropical Diseases, Ho Chi Minh city, Viet Nam; [5]Angkor Hospital for Children, Siem Reap, Cambodia; [6]University of Malaya Medical Centre, Kuala Lumpur, Malaysia; [7]Hospital Nacional de Niños Benjamin Bloom, San Salvador, El Salvador; [8]Centre for Tropical Medicine and Global health, Nuffield Department of Clinical Medicine, University of Oxford, Oxford, United Kingdom; [9]Institute for Vector-Borne Disease, Monash University, Clayton, Australia; [10]Section Clinical Tropical Medicine, Department for Infectious Diseases, Heidelberg University Hospital, Heidelberg, Germany; [11]Heidelberg Institute of Global Health (HIGH), Heidelberg University Hospital, Heidelberg, Germany; [12]Independent consultant, Issaquah, United States; [13]Consultant, Intellectual Ventures, Global Good Fund, Bellevue, United States; [14]UNICEF/UNDP/World Bank/WHO Special Programme for Research and Training in Tropical Diseases, World Health Organization, Geneva, Switzerland

\*For correspondence:
vuongnl@oucru.org (NLV);
syacoub@oucru.org (SY)

## Abstract

**Background:** Early identification of severe dengue patients is important regarding patient management and resource allocation. We investigated the association of 10 biomarkers (VCAM-1, SDC-1, Ang-2, IL-8, IP-10, IL-1RA, sCD163, sTREM-1, ferritin, CRP) with the development of severe/ moderate dengue (S/MD).
**Methods:** We performed a nested case-control study from a multi-country study. A total of 281 S/ MD and 556 uncomplicated dengue cases were included.
**Results:** On days 1–3 from symptom onset, higher levels of any biomarker increased the risk of developing S/MD. When assessing together, SDC-1 and IL-1RA were stable, while IP-10 changed the association from positive to negative; others showed weaker associations. The best combinations associated with S/MD comprised IL-1RA, Ang-2, IL-8, ferritin, IP-10, and SDC-1 for children, and SDC-1, IL-8, ferritin, sTREM-1, IL-1RA, IP-10, and sCD163 for adults.
**Conclusions:** Our findings assist the development of biomarker panels for clinical use and could improve triage and risk prediction in dengue patients.

**Funding:** This study was supported by the EU's Seventh Framework Programme (FP7-281803 IDAMS), the WHO, and the Bill and Melinda Gates Foundation.

## Introduction

Dengue is the most common arboviral disease to affect humans globally. In 2019, the World Health Organization (WHO) identified dengue as one of the top 10 threats to global health (*World Health Organization, 2019*). Transmission occurs in 129 countries, with an estimated 3.9 billion people being at risk (*World Health Organization, 2020*). Over the last two decades, the number of reported cases per year has increased more than eight-fold (*World Health Organization, 2020*), and in 2020 the annual number of dengue virus (DENV) infections was estimated to be 105 million, with 51 million cases being clinically apparent (*Cattarino et al., 2020*). With climate change, increased travel and urbanization, this rise is forecasted to continue over the coming decades (*Whitehorn and Yacoub, 2019*; *Yacoub et al., 2011*). Despite the large disease burden, there is still no specific treatment for dengue, and the only licensed vaccine is recommended only in individuals with earlier dengue infection (*Redoni et al., 2020*).

In many dengue-endemic settings, seasonal epidemics can rapidly overwhelm fragile health systems. Although most symptomatic dengue infections are self-limiting, a small proportion of patients develop complications, most of which manifest at around 4–6 days from symptom onset. Thus, large numbers of patients require regular assessments to identify complications should they arise. The accurate and early identification of such patients, particularly within the first 3 days of illness in the febrile phase, should allow for appropriate care to be provided and potentially increase health system effectiveness. Although the 2009 WHO dengue guidelines set out specific warning signs for use in patient triage, utility of these guidelines at identifying those at risk for complications remains limited (*Morra et al., 2018*).

The pathogenesis of dengue involves a complex interplay between viral factors and the host response. It is hypothesized that an excessive immune response acting through inflammatory mediators can lead to the observed manifestations of bleeding, shock, and organ dysfunction. Studies have shown that in secondary infections, adaptive immune activation can result in high circulating levels of plasma cytokines and chemokines (*Katzelnick et al., 2017*; *Midgley et al., 2011*; *Screaton et al., 2015*). Binding of viral NS1 protein onto endothelial cells can act in concert with vasoactive substances, cytokines, and chemokines, to result in endothelial activation and glycocalyx disruption, and these processes likely underlie the increased vascular permeability and coagulopathy (*McBride and Khanh Lam, 2020*; *Modhiran et al., 2015*; *St John et al., 2013*).

The role of blood biomarkers in predicting severe outcomes has been investigated in many studies, but mostly at later time-points or at hospital admission and many of these biomarkers either peak too late in the disease course or have too short a half-life to be clinically useful (*Ab-Rahman et al., 2016*; *John et al., 2015*; *Oliveira et al., 2017*; *Puerta-Guardo et al., 2019*; *Rathore et al., 2020*; *Robinson and Einav, 2020*; *S S et al., 2017*; *Soo et al., 2017*; *Vasey et al., 2020*; *Yacoub et al., 2017*; *Yacoub et al., 2016b*; *Yong et al., 2017*). Acknowledging these characteristics, we selected 10 candidate biomarkers from the vascular, immunological, and inflammatory pathways with good evidence supporting their involvement in the pathogenesis of dengue infection – focusing on those likely to be increased early in the disease course. We included vascular cell adhesion molecule-1 (VCAM-1), syndecan-1 (SDC-1), and angiopoietin-2 (Ang-2) because they represent endothelial activation and glycocalyx integrity (*Han et al., 2019*; *Mapalagamage et al., 2020*; *Suwarto et al., 2017*; *Yacoub et al., 2016a*). For markers of immune activation, we measured interleukin-8 (IL-8) and interferon gamma-induced protein-10 (IP-10) as these are associated with disease severity (*Oliveira et al., 2017*; *Pandey et al., 2015*; *Rathakrishnan et al., 2012*), and IL-1 receptor antagonist (IL-1RA), soluble cluster of differentiation 163 (sCD163), and soluble triggering receptor expressed on myeloid cells-1 (sTREM-1) as these are activation markers of monocytes and macrophages, the major targets for dengue replication (*Ab-Rahman et al., 2016*; *John et al., 2015*; *S S et al., 2017*). For markers of general inflammation, we included ferritin and C-reactive protein (CRP) (*Ab-Rahman et al., 2016*; *Finkelstein et al., 2020*; *Mukherjee and Tripathi, 2020*; *Soundravally et al., 2015*; *Vuong et al., 2020*).

The aims of this study were: (1) to investigate the association of these ten biomarkers with development of more severe dengue outcomes, (2) to find the best combination of biomarkers associated with more severe dengue outcomes. The results of the second aim could help in developing multiplex panels for use in outpatient settings to rapidly identify patients who require hospitalization.

## Materials and methods

### Study design

We conducted a nested case-control study using the samples and clinical information from a large multi-country observational study named 'Clinical evaluation of dengue and identification of risk factors for severe disease' (IDAMS study, NCT01550016) (*Jaenisch et al., 2016*). The IDAMS study and the blood sample analysis were approved by the Scientific and Ethics Committees of all study sites (Hospital for Tropical Diseases [Ho Chi Minh City, Vietnam] Ref No 03/HDDD-05/01/2018; Angkor Hospital for Children [Siem Reap, Cambodia] Ref No 0146/18-AHC; University of Malaya Medical Centre [Kuala Lumpur, Malaysia] Ref No 201865–6361) and by the Oxford Tropical Research Ethics Committee (OxTREC Ref No 502–18). There were 7428 participants in eight countries across Asia and Latin America enrolled in the IDAMS study. Patients were eligible for inclusion if they were aged 5 years or older, had fever or history of fever for less than 72 hr, and had symptoms consistent with dengue, with no features strongly suggestive of another disease. Participants were followed daily with a standard schedule of clinical examination and blood samples. Individual management (including hospitalization) was in accordance with routine practice at each study site. All diagnostic samples were processed and stored following specific protocols, and later transferred to designated sites for diagnostic testing in order to ensure consistency. Laboratory-confirmed dengue was defined by a positive reverse transcriptase polymerase chain reaction (RT-PCR) or a positive NS1 enzyme-linked immunosorbent assay (ELISA) result. Immune status was classified based on capture IgG results on paired samples. A probable primary infection was defined by two negative IgG results on two consecutive specimens taken at least 2 days apart, with at least one specimen obtained during the convalescent phase (after illness day 5). A probable secondary infection was defined by a positive IgG result identified during either or both the febrile and convalescent phases. In all other cases with the absence of suitable specimens at the appropriate time points immune status was classified as inconclusive. Each participant was given an overall severity grade (severe, moderate, or uncomplicated dengue), using all available information and a grading system in line with current guidelines and recommendations to classify clinical endpoints in dengue clinical trials (*Tomashek et al., 2018*).

### Study population

Of the 2694 laboratory-confirmed dengue cases in the IDAMS study, 38 and 266 cases were classified as severe and moderate dengue respectively. For this study, we selected all severe and moderate cases from five study sites in four countries (Vietnam, Cambodia, Malaysia, and El Salvador), as residual plasma from these countries' sample sets was available at the Oxford University Clinical Research Unit (OUCRU) in Ho Chi Minh City, Vietnam. For the control group, we selected patients with uncomplicated dengue with similar geographic and demographic characteristics at a 2:1 ratio. In total 281 cases and 556 controls were included in the analysis (*Figure 1*).

### Laboratory evaluation (details in Appendix 1)

The biomarkers were measured at two time points: at enrollment (illness day 1–3) and after recovery (day 10–31 post-symptom onset), if available. Eight biomarkers (CRP and ferritin excepted) were combined in a premixed magnetic bead panel (Cat No. LXSAHM; R and D). CRP was measured using a separate commercial magnetic bead panel (Cat. No. HCVD3MAG-67K; EMD Millipore Corporation). These panels were analyzed using the Luminex200 analyzer with the Luminex calibration (Cat. No. LX200-CAL-K25) and verification kits (Cat. No. LX200-CON-K25). Ferritin was measured using the Human Ferritin ELISA kit (Cat. No. ARG80501, Arigo). All tests were done according to the manufacturer's specifications.

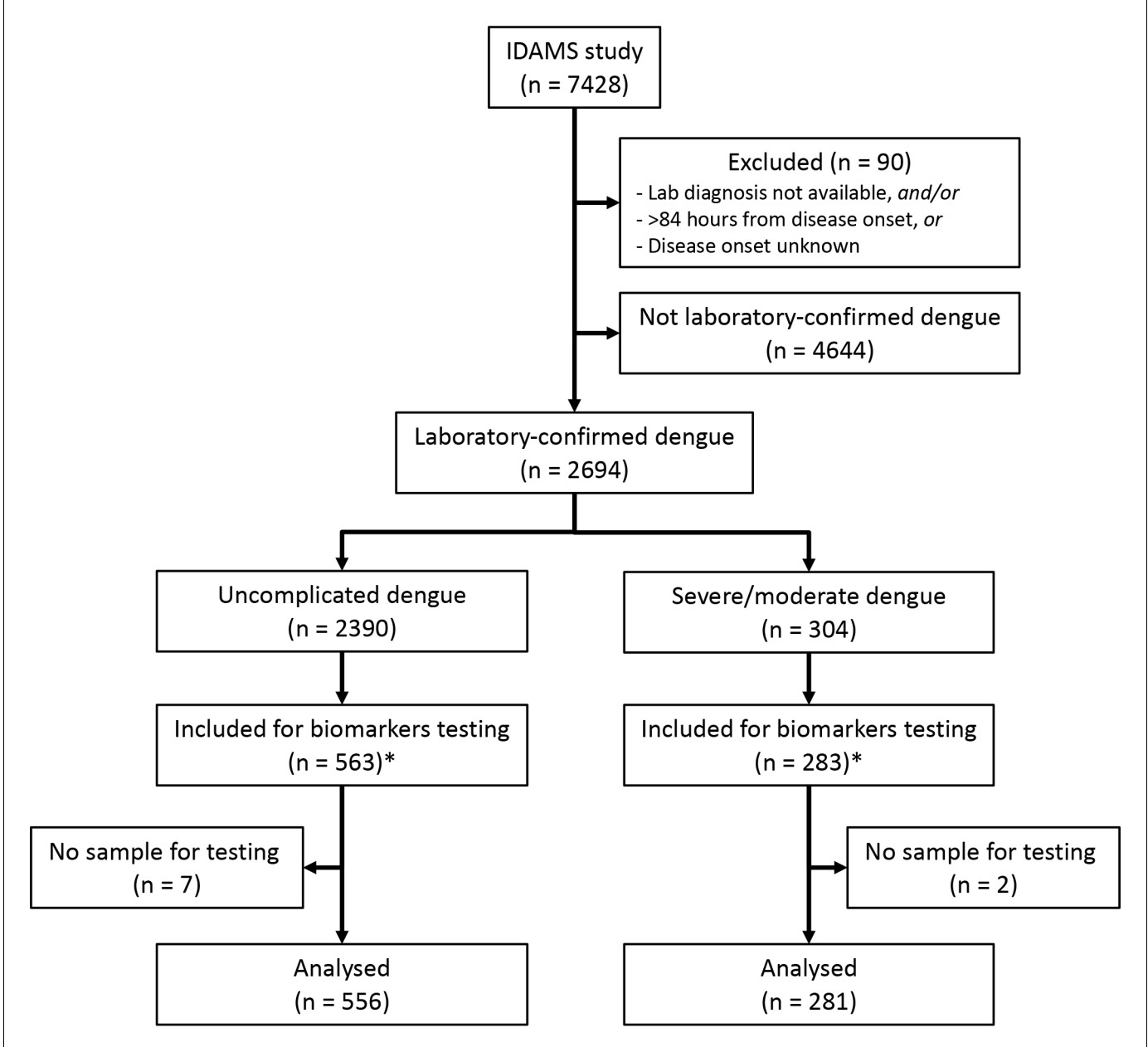

**Figure 1.** Study flowchart. *The IDAMS study was performed in eight countries across Asia and Latin America. For this study, we selected cases in four countries (Vietnam, Cambodia, Malaysia, and El Salvador) as the blood samples were stored at the laboratory of the Oxford University Clinical Research Unit in Ho Chi Minh City, Vietnam.

## Study endpoints (details in Appendix 2)

The primary endpoint was combined severe and moderate dengue (S/MD), defined by the development of severe or moderate grades of any of the following – plasma leakage, haemorrhage, or organ impairment (including neurologic, hepatic, or cardiac involvement) (*Appendix 2—table 1*). We combined severe and moderate dengue to form the primary endpoint (S/MD) as severe dengue events were rare; this combined endpoint is relevant to clinical practice since the moderate group is likely to develop complications and therefore may also require medical intervention and hospitalization. We studied three secondary endpoints: severe dengue alone, severe dengue or dengue with warning signs according to the 2009 WHO classification, and hospitalization. These endpoints were selected as they also reflect the disease burden and severity and are generalizable across different

**Table 1.** Summary of clinical data by primary outcome.

| | All patients | | Children | | Adults | |
|---|---|---|---|---|---|---|
| | Uncomplicated dengue (N = 556) | Severe/moderate dengue (N = 281) | Uncomplicated dengue (N = 337) | Severe/moderate dengue (N = 127) | Uncomplicated dengue (N = 219) | Severe/moderate dengue (N = 154) |
| **Country, n (%)** | | | | | | |
| - Cambodia | 39 (7) | 30 (11) | 37 (11) | 29 (23) | 2 (1) | 1 (1) |
| - El Salvador | 23 (4) | 18 (6) | 23 (7) | 18 (14) | 0 (0) | 0 (0) |
| - Malaysia | 58 (10) | 29 (10) | 3 (1) | 1 (1) | 55 (25) | 28 (18) |
| - Vietnam | 436 (78) | 204 (73) | 274 (81) | 79 (62) | 162 (74) | 125 (81) |
| Age (years), *median (1st, 3rd quartiles)* | 12 (9, 22) | 16 (10, 24) | 10 (8, 12) | 10 (7, 12) | 26 (20, 34) | 22 (18, 30) |
| Gender male, n (%) | 299 (54) | 170 (60) | 173 (51) | 80 (63) | 126 (58) | 90 (58) |
| **Illness day at enrolment, n (%)** | | | | | | |
| - 1 | 91 (16) | 49 (17) | 57 (17) | 25 (20) | 34 (16) | 24 (16) |
| - 2 | 260 (47) | 130 (46) | 156 (46) | 52 (41) | 104 (47) | 78 (51) |
| - 3 | 205 (37) | 102 (36) | 124 (37) | 50 (39) | 81 (37) | 52 (34) |
| **Serotype, n (%)** | | | | | | |
| - DENV-1 | 228 (41) | 121 (43) | 161 (48) | 61 (48) | 67 (31) | 60 (39) |
| - DENV-2 | 74 (13) | 47 (17) | 22 (7) | 16 (13) | 52 (24) | 31 (20) |
| - DENV-3 | 59 (11) | 29 (10) | 43 (13) | 18 (14) | 16 (7) | 11 (7) |
| - DENV-4 | 161 (29) | 70 (25) | 91 (27) | 26 (20) | 70 (32) | 44 (29) |
| - Unknown | 34 (6) | 14 (5) | 20 (6) | 6 (5) | 14 (6) | 8 (5) |
| **Immune status, n (%)** | | | | | | |
| - Probable primary | 124 (22) | 41 (15) | 86 (26) | 15 (12) | 38 (17) | 26 (17) |
| - Probable secondary | 355 (64) | 218 (78) | 202 (60) | 100 (79) | 153 (70) | 118 (77) |
| - Inconclusive | 77 (14) | 22 (8) | 49 (15) | 12 (9) | 28 (13) | 10 (6) |
| Obesity*, n (%) | 78 (14) | 29 (10) | 62 (18) | 19 (15) | 16 (7) | 10 (6) |
| Diabetes, n (%) | 4 (1) | 1 (0) | 0 (0) | 0 (0) | 4 (2) | 1 (1) |
| **WHO 2009 classification, n (%)** | | | | | | |
| - Mild dengue | 266 (48) | 49 (17) | 168 (50) | 17 (13) | 98 (45) | 32 (21) |
| - Dengue with warning signs | 288 (52) | 186 (66) | 169 (50) | 81 (64) | 119 (54) | 105 (68) |
| - Severe dengue | 0 (0) | 43 (15) | 0 (0) | 27 (21) | 0 (0) | 16 (10) |
| - Unknown | 2 (0) | 3 (1) | 0 (0) | 2 (2) | 2 (1) | 1 (1) |
| Hospitalization, n (%) | 175 (31) | 161 (57) | 127 (38) | 83 (65) | 48 (22) | 78 (51) |

*Obesity is defined as body mass index of higher than 30 kg/m$^2$ (for patients of older than 18 years) or two standard deviations of the median of body mass index for age (for patients of 18 years or below). WHO: World Health Organization.

settings. The decision to hospitalize was based only on clinical judgement and local guidelines particular to each study site, without use of any biomarker information.

## Statistical analysis (details in Appendix 3)

Plasma levels of all biomarkers were transformed to the base-2 logarithm (log-2) before analysis as a right skewed distribution was apparent. We used a logistic regression model for all endpoints. We investigated the non-linear effects of all biomarkers and age on the endpoints, using restricted cubic splines with three knots at the 10th, 50th, and 90th percentiles.

For the first aim, that is to investigate the association of all biomarkers with the primary and secondary endpoints, we performed two different analyses: (1) fitting models for each biomarker separately ('single models') and (2) fitting models including all biomarkers together ('global models'). In

the 'single models' for a particular biomarker, only that biomarker along with age and their interaction were included, whereas in the 'global models' all the biomarkers along with their interactions with age were included. We performed the 'global model' in order to investigate the influence of the biomarkers when considering them together and this was also the initial step to develop models for the second aim. Results are reported as odds ratio (OR) and presented graphically.

For the second aim to find the best combination of biomarkers associated with the primary endpoint, we built upon the results from the first aim to fit separate models for children and adults (<15 versus ≥15 years of age), as differences were apparent by age. We used variable selection based on the 'best subset' approach (*Hastie et al., 2017*; *Hocking and Leslie, 1967*). Briefly, this approach screened all possible combinations of biomarkers and selected the best based on the Akaike information criterion (AIC). We chose AIC as a ranking measurement because it quantifies goodness of fit, while guarding against over-fitting. The marker combination with the lowest AIC was taken as the best. From an 'initial model' including all biomarkers, we determined the best general combination and the best combinations of 2, 3, 4, and 5 biomarkers. We then performed a bootstrap procedure to check the robustness (stability) of the selected models. For this we resampled 1000 times with replacement from the original dataset. For each of these 1000 bootstrap samples, we performed the 'best subset' procedure similar to above to determine the best combination. We calculated the selection frequency of each marker combination over the 1000 samples. The frequency of the combination that was selected when using the original dataset in relation to the other combinations characterizes robustness of the selection.

We carried out several sensitivity analyses. First, we fitted the single and global models taking into account potential differences between serotypes by including serotype variable along with its interaction with the biomarkers. Second, we included viremia (viral RNA measured by RT-PCR) levels as an additional biomarker and performed the single model, global model and best subset procedure. Higher viremia levels have been associated with worse disease outcomes; however, viral load was not considered in the main analysis as the focus was on host markers with the potential for combining in a biomarker rapid test.

All analyses were done using the statistical software R version 3.6.3 (*R Core Development Team, 2020*) and the packages 'rms' (*Harrell Jr, 2019*), 'MuMIn' (*Bartoń, 2020*) and 'ggplot2' (*Wickham, 2016*). The code is available on GitHub (*Vuong, 2021b*; copy archived at swh:1:rev: 847d8e0f564eeb3f075b443205fb3384598bc2b4).

## Results

### Patient characteristics

The majority of the patients were from Vietnam (640 cases, 76%). Median (1st, 3rd quartiles) age of the case and control groups were 12 (9, 22) and 16 (10, 24) years. Among the S/MD group, 127 cases (45%) were children and 154 cases (55%) were adults. Male gender was predominant (60% and 54% in the case and control groups respectively). Serotype distribution was similar between the S/MD and control groups, with DENV-1 predominating (42%), particularly in children (48%). Host immune status however differed: there was a higher proportion of secondary infections in the S/MD group compared with controls (78% versus 64%, respectively) and this was consistent in both children and adults. The S/MD had a slightly lower percentage of obese patients than the control group (10% versus 14%). As expected, hospitalization was more common in the S/MD group (57% versus 31%) (*Table 1*). Overall, 38 patients developed severe dengue, most were severe plasma leakage (33/38 cases, 87%) and 29/38 (76%) were children. Most of the moderate dengue cases were plasma leakage and/or hepatic involvement (*Appendix 4—table 1*).

### Biomarker levels

On average, the patients who progressed to S/MD had higher levels of the biomarkers in both children and adult patients, both at enrollment and at follow-up (*Figure 2*, *Appendix 4—table 2*). For most individuals, the levels of five biomarkers (VCAM-1, IL-8, IP-10, IL-1RA, and CRP) decreased between enrollment and follow-up, whereas SDC-1 increased slightly and the other markers showed no clear trends (*Appendix 4—figure 1*). In some of the cases the biomarkers did not return to normal at convalescence. Moderate-to-strong positive correlations were evident for some markers, in

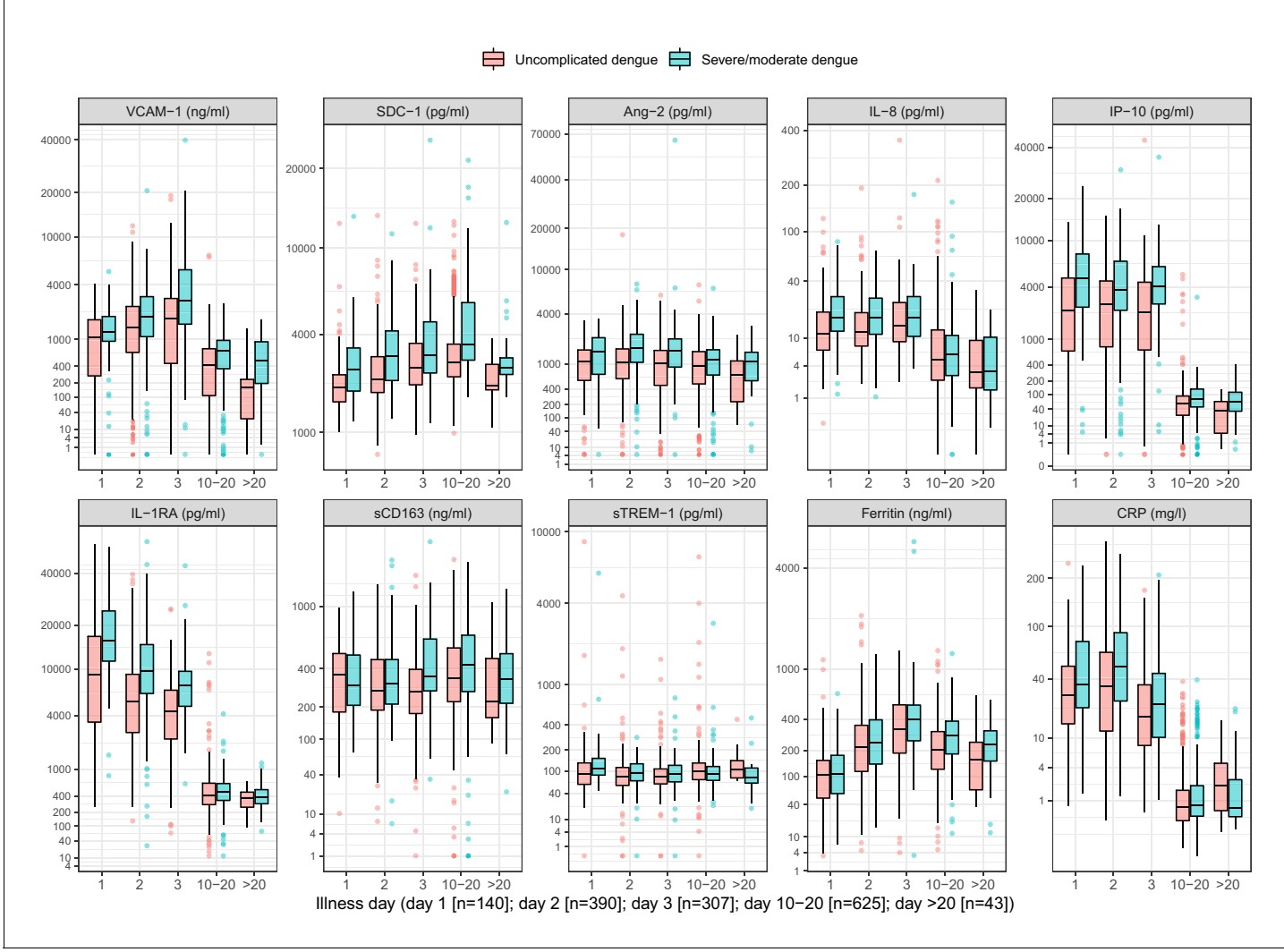

**Figure 2.** Biomarker levels by groups. VCAM-1: vascular cell adhesion molecule-1; SDC-1: syndecan-1; Ang-2: angiopoietin-2; IL-8: interleukin-8; IP-10: interferon gamma-induced protein-10; IL-1RA: interleukin-1 receptor antagonist; sCD163: soluble cluster of differentiation 163; sTREM-1: soluble triggering receptor expressed on myeloid cells-1; CRP: C-reactive protein. Y-axes are transformed using the fourth root transformation.

particular IP-10 and IL-1RA, and IP-10 and VCAM-1, both with Spearman's rank correlation coefficients above 0.6 (*Appendix 4—figure 2*).

## Associations between biomarker levels and the endpoints

In the single models, higher levels of each biomarker on illness days 1, 2, or 3 increased the risk of developing S/MD, with the exception of ferritin in adults where there was a downward trend at higher values (*Figure 3*, *Table 2*). We observed differences between children and adults for several biomarkers, the most pronounced being SDC-1, IL-8, ferritin, and IL-1RA. Associations between SDC-1 and IL-8 and the S/MD endpoint were stronger in adults than children, while the effects of IL-1RA and ferritin were stronger in children than adults.

In the global model there were some differences compared to the single models (*Figure 3*, *Table 2*). The biomarkers SDC-1 and IL-1RA were the most stable relative to the single models for both children and adults. However, for IP-10 the trend of the association with S/MD changed from positive to negative in both children and adults. In children, VCAM-1 changed the trend from positive to weakly negative and IL-8 changed the trend from weakly positive to negative. Other biomarkers showed weaker associations with the endpoint in the global model based on the ORs. In

**Table 2.** Results from models for the primary endpoint (severe or moderate dengue).

| | Single models | | | | Global model | | | |
|---|---|---|---|---|---|---|---|---|
| | Children OR (95% CI) | Adults OR (95% CI) | $P_{overall}$ | $P_{interaction}$ | Children OR (95% CI) | Adults OR (95% CI) | $P_{overall}$ | $P_{interaction}$ |
| VCAM-1 (ng/ml) | | | <0.001 | 0.715 | | | 0.441 | 0.213 |
| - 1636 vs 818 | 1.20 (1.04–1.38) | 1.35 (1.15–1.58) | | | 0.90 (0.73–1.10) | 1.22 (0.96–1.57) | | |
| - 3272 vs 1636 | 1.25 (1.02–1.53) | 1.48 (1.19–1.85) | | | 0.87 (0.66–1.15) | 1.30 (0.93–1.80) | | |
| SDC-1 (pg/ml) | | | <0.001 | 0.088 | | | 0.002 | 0.588 |
| - 2519 vs 1260 | 2.67 (1.31–5.43) | 3.33 (1.32–8.42) | | | 2.03 (0.77–5.34) | 5.11 (1.56–16.78) | | |
| - 5039 vs 2519 | 1.71 (1.18–2.47) | 3.71 (2.09–6.58) | | | 1.76 (0.98–3.14) | 2.52 (1.17–5.42) | | |
| Ang-2 (pg/ml) | | | <0.001 | 0.524 | | | 0.039 | 0.068 |
| - 1204 vs 602 | 1.64 (1.39–1.94) | 1.51 (1.26–1.82) | | | 1.67 (1.23–2.25) | 1.01 (0.74–1.38) | | |
| - 2409 vs 1204 | 2.21 (1.58–3.10) | 2.00 (1.40–2.85) | | | 1.95 (1.25–3.05) | 1.01 (0.65–1.57) | | |
| IL-8 (pg/ml) | | | <0.001 | <0.001 | | | <0.001 | <0.001 |
| - 14 vs 7 | 1.42 (1.05–1.91) | 2.18 (1.47–3.24) | | | 0.91 (0.63–1.34) | 1.69 (1.05–2.71) | | |
| - 28 vs 14 | 0.99 (0.78–1.25) | 2.33 (1.63–3.33) | | | 0.53 (0.36–0.77) | 2.05 (1.34–3.13) | | |
| IP-10 (pg/ml) | | | <0.001 | 0.984 | | | 0.206 | 0.630 |
| - 3093 vs 1546 | 1.46 (1.26–1.68) | 1.45 (1.21–1.73) | | | 0.94 (0.73–1.19) | 0.80 (0.57–1.12) | | |
| - 6186 vs 3093 | 1.68 (1.35–2.09) | 1.69 (1.29–2.22) | | | 1.08 (0.77–1.51) | 0.82 (0.52–1.29) | | |
| IL-1RA (pg/ml) | | | <0.001 | 0.082 | | | <0.001 | 0.032 |
| - 6434 vs 3217 | 1.69 (1.42–2.03) | 1.48 (1.21–1.81) | | | 2.07 (1.52–2.84) | 1.45 (0.98–2.15) | | |
| - 12868 vs 6434 | 1.82 (1.46–2.27) | 1.70 (1.29–2.24) | | | 2.16 (1.53–3.05) | 1.47 (0.94–2.30) | | |
| sCD163 (ng/ml) | | | <0.001 | 0.551 | | | 0.217 | 0.341 |
| - 295 vs 147 | 1.57 (1.14–2.15) | 1.49 (1.13–1.98) | | | 1.40 (0.89–2.22) | 1.27 (0.84–1.91) | | |
| - 589 vs 295 | 1.46 (1.10–1.93) | 1.61 (1.09–2.37) | | | 1.21 (0.87–1.69) | 1.39 (0.89–2.18) | | |
| sTREM-1 (pg/ml) | | | 0.059 | 0.997 | | | 0.555 | 0.393 |
| - 85 vs 42 | 1.87 (1.23–2.84) | 1.79 (1.10–2.93) | | | 1.13 (0.70–1.81) | 1.21 (0.65–2.26) | | |
| - 169 vs 85 | 1.12 (0.91–1.38) | 1.12 (0.82–1.53) | | | 0.89 (0.65–1.21) | 0.61 (0.38–0.99) | | |
| Ferritin (ng/ml) | | | 0.042 | 0.054 | | | 0.008 | 0.002 |
| - 243 vs 122 | 1.18 (1.01–1.38) | 1.06 (0.89–1.27) | | | 1.30 (1.04–1.64) | 0.78 (0.61–0.99) | | |
| - 487 vs 243 | 1.26 (1.00–1.58) | 0.90 (0.66–1.23) | | | 1.22 (0.89–1.67) | 0.66 (0.44–1.00) | | |
| CRP (mg/l) | | | <0.001 | 0.031 | | | 0.184 | 0.138 |
| - 28 vs 14 | 1.26 (1.12–1.41) | 1.25 (1.03–1.52) | | | 1.08 (0.93–1.25) | 1.10 (0.85–1.44) | | |
| - 56 vs 28 | 1.13 (0.95–1.34) | 1.38 (1.11–1.71) | | | 0.93 (0.75–1.15) | 1.36 (1.02–1.81) | | |

$P_{overall}$ is derived from Wald test for the overall association of the biomarker with the endpoint; $P_{interaction}$ is from the test for the interaction between the biomarker and age. The odds ratios are estimated at age of 10 and 25 years, represented as children and adults respectively.

addition, the differences of the associations between children and adults were more marked, particularly for Ang-2, IL-8, and ferritin.

The sensitivity analysis showed that the association between the biomarkers and S/MD did not differ between DENV-1 and other serotypes (*Appendix 5—figure 1*; *Appendix 5—figure 2*; *Appendix 5—table 1*; *Appendix 5—table 2*). Similar patterns were observed in the various analyses related to the secondary endpoints, as described in detail in the Appendix 6 (*Appendix 6—figure 1*; *Appendix 6—figure 2*, *Appendix 6—table 1*; *Appendix 6—table 2*, *Appendix 6—table 3*).

## Best combinations of biomarkers associated with the primary endpoint

For children, the best subset that showed the clearest association with S/MD was the combination of the six markers IL-1RA, Ang-2, IL-8, ferritin, IP-10, and SDC-1 with an AIC of 465.9. This model was

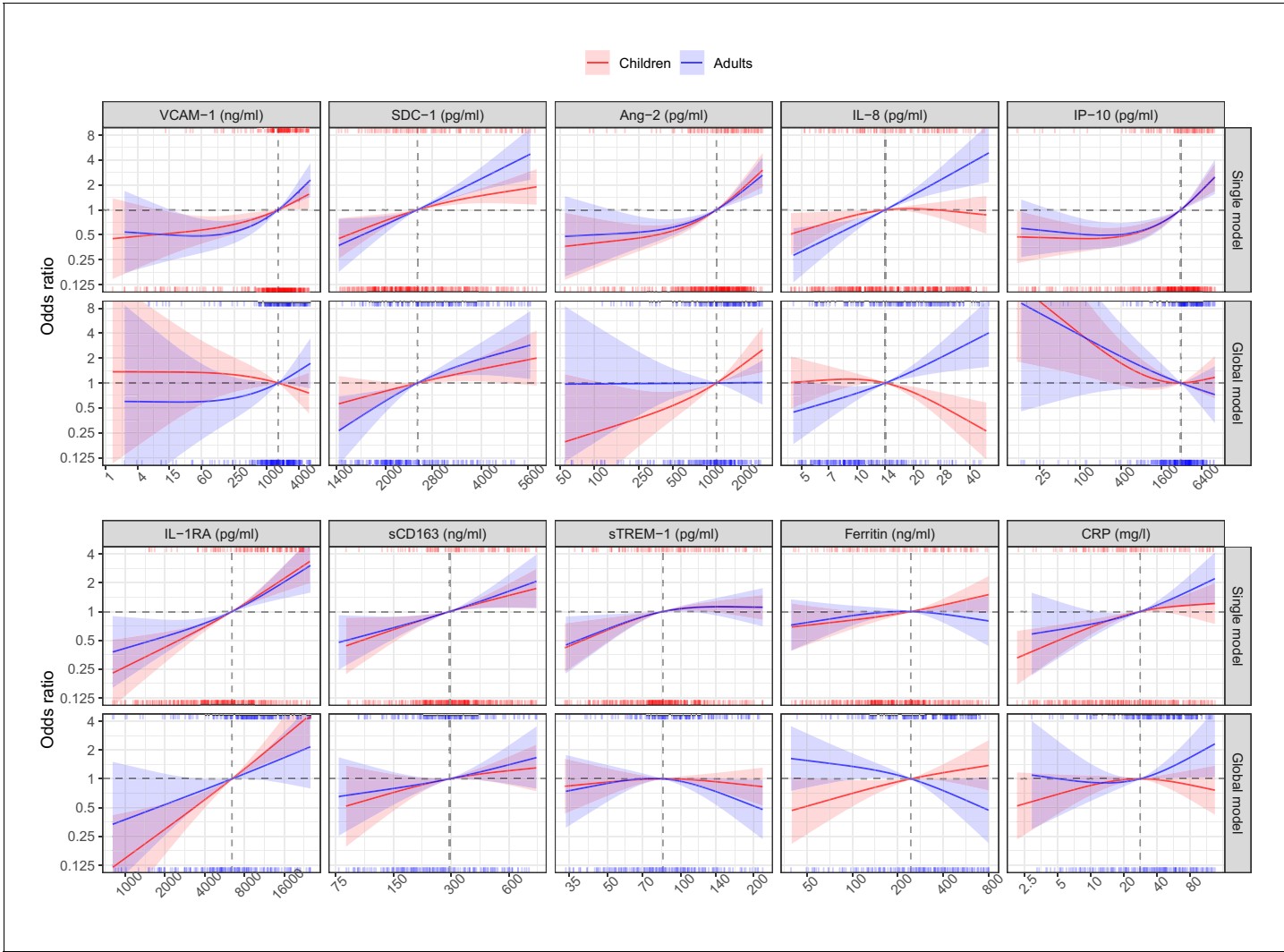

**Figure 3.** Results from models for the primary endpoint (severe or moderate dengue). The odds ratio of severe/moderate dengue (the red and blue lines) and 95% confidence interval (the red and blue regions) are estimated from multivariable logistic regression models allowing for a non-linear relation of log-2 of the biomarker level with severe/moderate dengue using restricted cubic splines. Each single model contains the corresponding biomarker, age and their interaction, while the global model contains all biomarkers and their interaction with age. The reference values for the odds ratios (where the odds ratio is equal to 1) are represented by the vertical gray dashed lines. They are chosen as the median of the biomarker levels of the whole study population (VCAM-1: 1636 ng/ml; SDC-1: 2519 pg/ml; Ang-2: 1204 pg/ml; IL-8: 14 pg/ml; IP-10: 3093 pg/ml; IL-1RA: 6434 pg/ml; sCD163: 295 ng/ml; sTREM-1: 85 ng/ml; ferritin: 243 ng/ml; and CRP: 28 mg/l). The x-axis represents biomarker levels; it is transformed using log-2 and its range truncated by the 5th and 95th percentiles of the biomarker levels of the whole study population. The rug plot on the x-axis represents the distribution of individual cases; the bottom rug plot represents the uncomplicated dengue cases and the top rug plot represents the severe/moderate dengue cases (children [<15 years of age] are in red and adults [≥15 years of age] are in blue). The red line and region represent children; results are shown for children at age of 10 years. The blue line and region represents adults; results are shown for adults at age of 25 years. VCAM-1: vascular cell adhesion molecule-1; SDC-1: syndecan-1; Ang-2: angiopoietin-2; IL-8: interleukin-8; IP-10: interferon gamma-induced protein-10; IL-1RA: interleukin-1 receptor antagonist; sCD163: soluble cluster of differentiation 163; sTREM-1: soluble triggering receptor expressed on myeloid cells-1; CRP: C-reactive protein.

selected most often in the bootstrap procedure, but was not highly robust (it was selected in 134 of the 1000 samples) (*Table 3*, *Appendix 7—table 1*). Over the 1000 samples, the six variables had an inclusion frequency ranging from 73.5% for SDC-1 to 100% for IL-1RA. The most important biomarkers in order were IL-1RA, Ang-2, IL-8, and ferritin (*Appendix 7—table 2*). The best combination of two biomarkers was IL-1RA and ferritin, the best of three added Ang-2, the best of four added IP-10, and the best of five added IL-8. The best combinations of two and five variables were most robust with a selection percentage of 43.7% and 44%. The best of five had almost the same AIC as

the best subset of six markers (467.6 versus 465.9) (*Table 3*). The coefficients of the selected biomarkers were similar to the initial model estimates (*Appendix 7—table 2*).

For adults, the best subset included the seven markers SDC-1, IL-8, ferritin, sTREM-1, IL-1RA, IP-10, and sCD163. This model was selected 79 times among 1000 bootstrap samples, but still was selected more often than the other models (*Table 4*, *Appendix 7—table 3*). Over the 1000 samples, the seven variables had a bootstrap inclusion frequency ranging from 59.1% for sCD163 to 99.2% for SDC-1. The three most important biomarkers in order were SDC-1, IL-8, and ferritin (*Appendix 7—table 4*). The best combination of two biomarkers included SDC-1 and IL-8, the best of three added ferritin, the best of four added IL-1RA, and the best of five added sTREM-1. The best combination of two was the most robust with a selection percentage of 56.7%, followed by the best of three variables (43.2%) (*Table 4*). The coefficients of the selected markers were also similar to the initial model estimates (*Appendix 7—table 4*).

In the sensitivity analysis, viremia was not selected in any of the best combinations for children, and the marker combinations remained the same as the main analysis. For adults, the best subset included five markers SDC-1, IL-8, ferritin, viremia and sCD163. The best of two and three were the same as the main analysis; viremia was selected in the best of four and five (*Appendix 8—figure 1*; *Appendix 8—table 1*; *Appendix 8—table 2*; *Appendix 8—table 3*).

## Discussion

This nested case-control study has shown that a range of endothelial, immune activation and inflammatory biomarkers measured during the early febrile phase of dengue are associated with progression to worse clinical outcomes in both children and adults. In children we found IL-1RA to have the most robust association with S/MD, whereas in adults we found SDC-1 and IL-8 to have the most robust association. For children, the best combination (ordered by robustness) included six biomarkers IL-1RA, Ang-2, IL-8, ferritin, IP-10, and SDC-1; for adults the best combination identified comprised seven biomarkers SDC-1, IL-8, ferritin, sTREM-1, IL-1RA, IP-10, and sCD163.

Our results add to the current literature on biomarkers in severe/moderate dengue compared with uncomplicated dengue, by including early time-points prior to the development of the severe manifestations, as well as providing data on the use of biomarker combinations, which takes into consideration the complex inflammatory-vascular pathogenesis of severe dengue. We observed that there were marked changes in the associations between individual biomarkers and outcomes when considering them together, while other biomarkers showed consistent associations. Particularly, the association of IP-10 with S/MD changed significantly from the single to global model, which may be because another biomarker in our model is a mediator or confounder of IP-10 in the pathway to the outcome. This could be IL-1RA as its association with S/MD was similar between the single and global model, and the correlation between IP-10 and IL-1RA was strong (Spearman's rank correlation coefficient was 0.75). Nonetheless, changing the direction of the association from the single to global model does not diminish the possibility of that biomarker being selected in the best combinations.

Our study also demonstrates some key differences between pediatric and adult dengue. Clinical phenotypes of dengue in children and adults differ, with children experiencing more shock and adults more organ impairment and bleeding, with distinct clinical management guidelines published by the WHO. Our results imply dengue pathogenesis may differ by age, with distinct combinations of immune-activation and vascular markers demonstrated between children and adults. Specifically, the association of IL-8 and ferritin differed between children and adults, which is likely to be due to the composite endpoint of severe and moderate dengue. As shown in the analysis of severe dengue alone (*Appendix 4—figure 1*, *Appendix 4—table 1*), the effects of IL-8 and ferritin were similar in children and adults, which suggests these biomarkers are still associated with severe disease in all age groups and that the difference is driven by the moderate dengue group. In addition, uncomplicated dengue in adults have higher ferritin levels compared to in children, with increasing age and chronic conditions in adults likely contributing to this observation. Hence patients' age should be considered when developing biomarker panels for dengue risk prediction.

The use of biomarker panels for the prediction of severe outcomes in dengue has been investigated in previous studies, using several statistical approaches (*Brasier et al., 2012*; *Conroy et al., 2015*; *Ju and Brasier, 2013*; *Lee et al., 2016*; *Pang et al., 2016*). However, because of small sample

**Table 3.** Best combinations of biomarkers associated with severe or moderate dengue for children.

| | Best of all combinations | Best combination of 2 variables | Best combination of 3 variables | Best combination of 4 variables | Best combination of 5 variables |
|---|---|---|---|---|---|
| **Variables** | | | | | |
| - VCAM-1 | | | | | |
| - SDC-1 | + | | | | |
| - Ang-2 | + | | + | + | + |
| - IL-8 | + | | | | + |
| - IP-10 | + | | | + | + |
| - IL-1RA | + | + | + | + | + |
| - sCD163 | | | | | |
| - sTREM-1 | | | | | |
| - Ferritin | + | + | + | + | + |
| - CRP | | | | | |
| AIC of the selected model | 465.9 | 484.7 | 480.0 | 473.7 | 467.6 |
| **Bootstrap results** | | | | | |
| - Model selection frequency, *n (%)* | 134 (13.4) | 437 (43.7) | 239 (23.9) | 317 (31.7) | 440 (44.0) |
| - Rank by selection frequency of the selected model | 1 | 1 | 2 | 1 | 1 |

VCAM-1: vascular cell adhesion molecule-1; SDC-1: syndecan-1; Ang-2: angiopoietin-2; IL-8: interleukin-8; IP-10: interferon gamma-induced protein-10; IL-1RA: interleukin-1 receptor antagonist; sCD163: soluble cluster of differentiation 163; sTREM-1: soluble triggering receptor expressed on myeloid cells-1; CRP: C-reactive protein; AIC: Akaike information criterion.

**Table 4.** Best combinations of biomarkers associated with severe or moderate dengue for adults.

| | Best of all combinations | Best combination of 2 variables | Best combination of 3 variables | Best combination of 4 variables | Best combination of 5 variables |
|---|---|---|---|---|---|
| **Variables** | | | | | |
| - VCAM-1 | | | | | |
| - SDC-1 | + | + | + | + | + |
| - Ang-2 | | | | | |
| - IL-8 | + | + | + | + | + |
| - IP-10* | + | | | | |
| - IL-1RA | + | | | + | + |
| - sCD163 | + | | | | |
| - sTREM-1 | + | | | | + |
| - Ferritin | + | | + | + | + |
| - CRP | | | | | |
| AIC of the selected model | 430.5 | 441.1 | 434.2 | 431.6 | 430.7 |
| **Bootstrap results** | | | | | |
| - Model selection frequency, *n (%)* | 79 (7.9) | 567 (56.7) | 432 (43.2) | 202 (20.2) | 161 (16.1) |
| - Rank by selection frequency of the selected model | 1 | 1 | 1 | 1 | 1 |

VCAM-1: vascular cell adhesion molecule-1; SDC-1: syndecan-1; Ang-2: angiopoietin-2; IL-8: interleukin-8; IP-10: interferon gamma-induced protein-10; IL-1RA: interleukin-1 receptor antagonist; sCD163: soluble cluster of differentiation 163; sTREM-1: soluble triggering receptor expressed on myeloid cells-1; CRP: C-reactive protein; AIC: Akaike information criterion.
*Variable is kept as non-linear effect using natural cubic splines with three knots.

size and differences in the biomarkers assessed, the associations found vary between studies and as yet there are no validated prognostic panels for dengue. Dengue cases are forecasted to increase over the next few decades and, given the limited healthcare resources available in many endemic settings, particularly during epidemics, there is an urgent need to develop innovative methods to rapidly identify patients likely to develop complications and require hospital care (*Rodriguez-Manzano et al., 2018*). Previously, we showed that CRP as a single biomarker was useful for early dengue diagnosis and risk identification, which is currently easy to use in all settings (*Vuong et al., 2020*). Recently, we also showed that higher plasma viremia was associated with increased dengue severity regardless of age, serotype and immune status of patients (*Vuong et al., 2021a*). However, future point-of-care testing could be improved by using a combination of biomarkers outlined in this study. Our results are applicable to the development of point-of-care panels capable of multiplex analysis and suited for use in outpatient settings for dengue prognosis, with scope for incorporation with innovative point-of-care technologies. To be more applicable by balancing model fit, robustness, and parsimony, we suggest the combination of five biomarkers IL-1RA, Ang-2, IL-8, ferritin, and IP-10 for children, and the combination of three biomarkers SDC-1, IL-8, and ferritin for adults to be used in practice. These combinations had a similar AIC with the best combinations (the difference of AIC was less than 5), but they required fewer number of biomarkers in a test panel. With the advent of novel technologies including microarray platforms and multiplex lateral flow assays, the cost is likely to come down in the future, allowing for wide-spread use in low-to-middle-income countries.

Methods of variable selection have been discussed previously but there remains no clear consensus regarding the best approach (*Heinze et al., 2018*; *Sauerbrei et al., 2020*). We adopted a data-driven 'best subset' approach which we think offers advantages over other methods, given the complexity of the biomarkers involved and their interactions. We also explored other approaches for variable selection (*Heinze et al., 2018*; *Piironen and Vehtari, 2017*; *Sauerbrei et al., 2020*) and the results were very similar to the best subset procedure (*Appendix 9—table 1*; *Appendix 9—table 2*).

Strengths of our study include the large sample size and use of a nested case-control dataset from a prospective multi-country cohort study with consistent data collection and standardized outcome definitions and laboratory methodologies. The biomarker panel we selected was guided by pathogenesis studies, focusing on pathways activated early in the disease course, thus ensuring clinical relevance.

There are some limitations in our study. One being we analysed the biomarkers at only one time-point in the early phase; limited financial resources did not allow us to evaluate the full range of biomarkers across the whole IDAMS population and at more time-points. Secondly, this study was not designed to build prediction models so we did not use a measure of predictive value as a criterion, which was motivated by the nested case-control design. Our findings need to be validated in new studies.

In conclusion, higher levels of the ten biomarkers (VCAM-1, SDC-1, Ang-2, IL-8, IP-10, IL-1RA, sCD163, sTREM-1, ferritin, and CRP), when considered individually, are associated with increased risk of adverse clinical outcomes in both children and adults with dengue. The best biomarker combination for children includes IL-1RA, Ang-2, IL-8, ferritin, IP-10, and SDC-1; for adults, SDC-1, IL-8, ferritin, sTREM-1, IL-1RA, IP-10, and sCD163 were selected. These findings serve to assist the development of biomarker panels to improve future triage and early assessment of dengue patients. This would aid not only individual patient management and facilitate healthcare allocation which would be of major public health benefit especially in outbreak settings, but could also serve as potential biological endpoints for dengue clinical trials.

## Acknowledgements

We acknowledge all the patients who took part in this study and the medical and nursing staff who looked helped in their management, at all the participating hospitals in Vietnam, Cambodia, Malaysia, and El Salvador. This work was supported by the European Union's Seventh Framework Programme for research, technological development and demonstration (grant FP7-281803 IDAMS; http://www.idams.eu; publication reference number IDAMS: 53), and by the World Health Organization, UNICEF/UNDP/World Bank/WHO Special Programme for Research and Training in Tropical

Diseases, and by the Bill and Melinda Gates Foundation Trust through The Global Good Fund I, LLC at Intellectual Ventures. The funders had no role in the study design, data collection and analysis, or preparation of the manuscript. The authors alone are responsible for the views expressed in this article and they do not necessarily represent the views, decisions or policies of the institutions with which they are affiliated.

## Additional information

### Competing interests

Damien Keng Yen Ming: reports receiving personal fees from the Wellcome Trust (grant number 215010/Z/18/Z), during the conduct of the study. Lucy Chai See Lum: reports receiving personal fees from ROCHE Advisory Board on Severe Dengue. Thomas Jaenisch: reports receiving personal fees as members of the ROCHE Advisory Board on Severe Dengue, outside the submitted work. Bridget A Wills: reports personal fees as a member of the Data Monitoring and Adjudication Committees for the Takeda dengue vaccine trials and as a member of the ROCHE Advisory Board on Severe Dengue, outside the submitted work. Ronald B Geskus: reports receiving personal fees from the Wellcome Trust (grant number 106680/Z/14/Z), during the conduct of the study. Sophie Yacoub: reports receiving personal fees as a member of the ROCHE Advisory Board on Severe Dengue, for work on Janssen Pharmaceuticals Advisory Board for Dengue Antiviral Development, and from the Wellcome Trust (grant number 106680/Z/14/Z). The other authors declare that no competing interests exist.

### Funding

| Funder | Grant reference number | Author |
| --- | --- | --- |
| European Union Seventh Framework Programme | FP7-281803 IDAMS | Thomas Jaenisch |
| World Health Organization | UNICEF/UNDP/ World Bank/WHO Special Programme for Research and Training in Tropical Diseases | Sophie Yacoub |
| Bill and Melinda Gates Foundation | The Global Good Fund I, LLC at Intellectual Ventures | Sophie Yacoub |
| Wellcome Trust | 106680/Z/14/Z | Ronald B Geskus Sophie Yacoub |
| Wellcome Trust | 215010/Z/18/Z | Damien Keng Yen Ming |

The funders had no role in study design, data collection and interpretation, or the decision to submit the work for publication.

### Author contributions

Nguyen Lam Vuong, Conceptualization, Software, Formal analysis, Visualization, Methodology, Writing - original draft, Writing - review and editing; Phung Khanh Lam, Conceptualization, Data curation, Software, Supervision, Investigation, Methodology, Writing - review and editing; Damien Keng Yen Ming, Conceptualization, Investigation, Writing - review and editing; Huynh Thi Le Duyen, Resources, Data curation, Software, Writing - review and editing; Nguyet Minh Nguyen, Ngoun Chanpheaktra, Lucy Chai See Lum, Ernesto Pleités, Conceptualization, Data curation, Investigation, Writing - review and editing; Dong Thi Hoai Tam, Conceptualization, Data curation, Investigation, Methodology, Writing - review and editing; Kien Duong Thi Hue, Data curation, Writing - review and editing; Nguyen VV Chau, Cameron P Simmons, Kerstin D Rosenberger, Conceptualization, Investigation, Methodology, Writing - review and editing; Thomas Jaenisch, Conceptualization, Funding acquisition, Investigation, Methodology, Writing - review and editing; David Bell, Nathalie Acestor, Christine Halleux, Piero L Olliaro, Conceptualization, Funding acquisition, Writing - review and editing; Bridget A Wills, Sophie Yacoub, Conceptualization, Supervision, Funding acquisition,

Investigation, Methodology, Writing - review and editing; Ronald B Geskus, Conceptualization, Supervision, Investigation, Methodology, Writing - review and editing

### Author ORCIDs

Nguyen Lam Vuong https://orcid.org/0000-0003-2684-3041
Phung Khanh Lam https://orcid.org/0000-0001-7968-473X
Damien Keng Yen Ming https://orcid.org/0000-0003-3125-6378
Nguyet Minh Nguyen https://orcid.org/0000-0002-5960-7849
Lucy Chai See Lum https://orcid.org/0000-0002-7452-4208
Cameron P Simmons http://orcid.org/0000-0002-9039-7392
David Bell https://orcid.org/0000-0002-7010-6340
Bridget A Wills https://orcid.org/0000-0001-9086-8804
Ronald B Geskus https://orcid.org/0000-0002-2740-3155

### Ethics

Human subjects: The study and the blood sample analysis were approved by the Scientific and Ethics Committees of all study sites (Hospital for Tropical Diseases [Ho Chi Minh City, Vietnam] Ref No 03/HDDD-05/01/2018; Angkor Hospital for Children [Siem Reap, Cambodia] Ref No 0146/18-AHC; University of Malaya Medical Centre [Kuala Lumpur, Malaysia] Ref No 201865-6361) and by the Oxford Tropical Research Ethics Committee (OxTREC Ref No 502-18).

### Decision letter and Author response

Decision letter https://doi.org/10.7554/eLife.67460.sa1
Author response https://doi.org/10.7554/eLife.67460.sa2

## Additional files

### Supplementary files

• Transparent reporting form

### Data availability

All data generated or analysed during this study have been deposited in the Oxford Research Archive (ORA) at https://doi.org/10.5287/bodleian:JN2wXDpjq and all code has been deposited on GitHub at https://github.com/Nguyenlamvuong/eLife_Biomarkers_Dengue_2021 copy archived at https://archive.softwareheritage.org/swh:1:rev:847d8e0f564eeb3f075b443205fb3384598bc2b4.

The following dataset was generated:

| Author(s) | Year | Dataset title | Dataset URL | Database and Identifier |
|---|---|---|---|---|
| Yacoub S | 2021 | Combination of inflammatory and vascular markers in the febrile phase of dengue is associated with more severe outcomes | https://doi.org/10.5287/bodleian:JN2wXDpjq | Oxford Research Archive, 10.5287/bodleian:JN2wXDpjq |

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

# Appendix 1

## Laboratory evaluation of the ten biomarkers

All research blood samples enrolled at different study sites were processed at site laboratories within one hour after drawn from participants with the same procedure. The blood samples were centrifuged at 500 g/min in 10 min at 4°C, collected plasma, and stored at −20°C. The specimens enrolled from international sites were transported on dry ice to the Oxford University Clinical Research Unit (OUCRU) laboratory by worldwide couriers. Biomarker levels were measured on these stored samples at two time-points: enrollment sample (illness day 1–3) and follow-up (day 10–31 post symptom onset) using the quantitative magnetic bead assays and enzyme-linked immunosorbent assays (ELISAs).

Eight biomarkers (VCAM-1, SDC-1, Ang-2, IL-8, IP-10, IL-1RA, sCD163, and sTREM-1) combined in a premixed magnetic bead panel (Cat No. LXSAHM; R and D) was selected to investigate these interested targets. A commercial kit used for CRP level measurements was Human Cardiovascular Disease (CVD) Magnetic Bead Panel 3 (Cat. No. HCVD3MAG-67K) produced by EMD Millipore Corporation. Each assay had two quality controls (low and high levels) which was acquired along with the standards and unknown specimens.

The xPonent 3.1 software installed in the Luminex200 analyzer was used to acquire and analyze the data of the magnetic luminex assays. The system was calibrated daily using the Luminex calibration (Cat. No. LX200-CAL-K25) and verification kits (Cat. No. LX200-CON-K25). Equipment settings included probe height adjustment, number of events for each analyte, sample size, gate settings, and bead set were followed as the kit's instruction. A background was set up using assay buffer instead of sample in all luminex assays.

As recommended by manufacturers, the magnetic bead assays were performed using samples with less than two freeze/thaw cycles and the samples after thawed completely were centrifuged to remove particles prior to use in the assays. The samples were diluted at the different dilutions in order to fall within the standard curve range in each assay.

The magnetic bead assays were designed in multiplex sandwich ELISAs. The magnetic microparticles pre-coated with specific antibodies were pipetted into the wells containing standards or controls or diluted samples and the immobilized antibodies bound the analytes of interest for two hours incubation at room temperature or overnight (16–18 hr) at 4°C on a shaker. Then, these analytes were detected specifically by a secondary biotinylated antibody cocktail during the next incubation. After being washed to remove any unbound antibody, the streptavidin-phycoerythrin conjugate was added to bind to the biotinylated antibody. Finally, the microparticles were re-suspended in buffer and read using the Luminex200 analyzer. The microparticles were re-suspended immediately prior to reading by shaking the plate for two minutes on the shaker.

Ferritin levels were measured separately using Human Ferritin ELISA kit (Cat. No. ARG80501, Arigo). This assay is a quantitative sandwich ELISA. An antibody specific for Ferritin was pre-coated onto a microplate. Standard or samples were pipetted into the wells and any Ferritin present was bound by the immobilized antibody. After washing away any unbound substances, a horseradish peroxidase (HRP) conjugated antibody specific for ferritin was added to each well and incubated. A substrate solution (3,3′,5,5′-tetramethylbenzidine [TMB]) was then added to the wells and color developed in proportion to the amount of ferritin bound in the initial step. The color development was stopped by addition of acid and the intensity of the color was measured by a wavelength of 450 nm. The concentration of ferritin in the sample was then determined by comparing the optical density of samples to the standard curve.

# Appendix 2

## Clinical endpoint definition

**Appendix 2—table 1.** Definition of severe and moderate dengue components.

| Endpoint | Definition |
|---|---|
| Severe plasma leakage | Dengue shock syndrome or respiratory distress due to plasma leakage |
| Moderate plasma leakage | Did not fulfill criteria for severe plasma leakage and had at least one of the following criteria: (1) maximum haematocrit change was 20% or more, and (2) having evidence of fluid accumulation |
| Severe bleeding | Any bleeding into a critical organ or required any blood transfusion of packed red cells or whole blood without pre-anaemia, or bleeding with complication |
| Moderate bleeding | Did not fulfill criteria for severe bleeding and had at least one of the following criteria: (1) severe bleeding by clinical judgement, (2) bleeding required any blood transfusion other than packed red cells or whole blood, (3) bleeding required other intervention (e.g. nasal packing, cross-match, etc.), and (4) receiving packed red cells or whole blood with pre-existing anaemia and with a consistent haemoglobin value |
| Severe neurologic involvement | Abnormal neurologic examination and neurologic involvement that resulted in death or ongoing sequelae that impaired daily function, or required intubation, shunting or intensive care |
| Moderate neurology involvement | Single convulsion without hospitalization or other complication |
| Severe hepatic involvement | Jaundice or coagulopathy or encephalopathy |
| Moderate hepatic involvement | Any alanine aminotransferase (ALT) or aspartate aminotransferase (AST) result of 400 IU/L or more |
| Severe other major organ failure | Creatine kinase or other enzymes (e.g. troponin) abnormalities and functional abnormalities (e.g. reduced cardiac ejection fraction less than 50% or new electrocardiogram [ECG] abnormalities) or required specific intervention (e.g. inotropic support) |
| Moderate other major organ failure | Troponin abnormalities alone or creatine kinase abnormalities without cardiac ejection fraction less than 50% |

## Appendix 3

### Statistical analysis

#### Treatment of values lower than the limit of detection

There were several biomarker values lower than the limit of detection (LOD): 9% for VCAM-1, 5% for Ang-2, 1% for IP-10 and sTREM-1,<1% for IL-8 and sCD163, and none for SDC-1, IL-1RA, ferritin, and CRP. All of them were set to the LOD and a dummy binary variable (Yes/No) was created for each biomarker to describe whether the value was lower than the LOD or not. In all models for the first aim (to investigate the association of biomarkers with clinical outcomes), for each biomarker that had values below the LOD, we included the binary variable '<LOD' as a covariate.

#### Analysis of the secondary endpoints: case-control setting

Since cases and controls were selected based on the primary endpoint, in the analyses of the secondary endpoints we lose the case-control distinction. We used inverse probability weighting (IPW) to correct for different inclusion probabilities between controls and cases in our data set (*Tchetgen Tchetgen, 2014*; *Schifano, 2019*): the weight of all cases was 1, while the weight of the controls was the inverse of the inclusion probability in each country (Vietnam: 1301/436; Cambodia: 272/39; Malaysia: 230/58; and El Salvador: 288/23). A robust (sandwich) estimate of the standard error was used for estimating 95% confidence intervals. For the severe dengue (SD) endpoint, the non-linear effect was not considered because of the low number of events.

#### Analysis to find the best combination of biomarkers to predict the primary endpoint (aim #2)

The results from the 'single models' and 'global model' in the first aim showed that the association between the biomarkers and the primary endpoint differed by age. We therefore performed the analysis separately for children (<15 years of age) and adults ($\geq$15 years of age). The procedure was done in two steps. In step #1 we built an 'initial model' including all biomarkers, but possibly with less flexible structure than the global model. In step #2 we determined the best combination of biomarkers from the initial models defined in step #1.

Step #1: As the number of primary endpoint events was limited (127 in children and 154 in adults), we tried to keep the events-per-variable (EPV) at more than 10 by including only important terms (all the ten biomarkers but only some of the non-linear trends and binary variables that represent values < LOD) (*Heinze et al., 2018*). For each biomarker, we fitted and compared four logistic regression models:

1. model with the biomarker with a linear effect as the only covariate:
   $\text{logit}(Y) = \alpha + \beta*X$
2. model with the biomarker with a non-linear effect using restricted cubic splines (as in the single and global model):
   $\text{logit}(Y) = \alpha + \beta_1*\text{spline}(X)_1 + \beta_2*\text{spline}(X)_2$
3. model with the biomarker with a linear effect and the dummy binary variable for value <LOD:
   $\text{logit}(Y) = \alpha + \beta*X + \gamma*(X < LOD)$
4. model with the biomarker with a non-linear effect and the dummy binary variable for value <LOD:
   $\text{logit}(Y) = \alpha + \beta_1*\text{spline}(X)_1 + \beta_2*\text{spline}(X)_2 + \gamma*(X < LOD)$

Y is the primary endpoint, X is a biomarker

We calculated and compared the Akaike information criterion (AIC) of these four models and included the non-linear effect and/or additional binary variable of values < LOD only if it had the

lowest AIC and this value was at least five lower than for model (1). The AICs of these models are summarized in the table below:

| Biomarker | Children | | | | Adults | | | |
|---|---|---|---|---|---|---|---|---|
| | model1 | model2 | model3 | model4 | model1 | model2 | model3 | model4 |
| VCAM-1 | **530.8** | 531.3 | 530.6 | 532.6 | **499.8** | 496.7 | 501.3 | 496.4 |
| SDC-1 | **537.9** | 538.0 | - | - | **459.5** | 461.0 | - | - |
| Ang-2 | **511.0** | 509.6 | 512.5 | 509.7 | **493.5** | 490.8 | 493.3 | 492.8 |
| IL-8 | **548.5** | 549.0 | 548.1 | 546.8 | **457.7** | 457.8 | 456.5 | 457.8 |
| IP-10 | **521.2** | 517.2 | 523.0 | 518.3 | 500.7 | **492.6** | 502.5 | 494.5 |
| IL-1RA | **492.9** | 494.9 | - | - | **493.5** | 494.2 | - | - |
| sCD163 | **531.9** | 533.2 | 531.9 | 533.2 | **497.6** | 499.3 | 499.5 | 501.1 |
| sTREM-1 | **545.7** | 544.2 | 545.6 | 545.4 | **507.8** | 509.5 | 507.0 | 505.0 |
| Ferritin | **542.6** | 544.4 | - | - | **509.3** | 505.2 | - | - |
| CRP | **536.7** | 536.4 | - | - | **505.1** | 506.8 | - | - |

The selected models are in bold face.

The initial model for children included all biomarkers as a linear term without any binary variable for values < LOD, and the final initial model for adults included IP-10 with a non-linear term and all the other biomarkers with a linear term, again without any binary variable for values < LOD. The number of parameters of the initial model was 10 and 11 for children and adults; the EPV was then 12.7 and 14 respectively.

Step #2: From the 'initial model', we performed several approaches to find the best combination of biomarkers associated with the primary endpoint.

The primary method was the 'best subset' approach, in which all possible subsets of biomarkers ($2^{10}$ = 1024 subsets) were evaluated and compared via the AIC. The subset with the lowest AIC was selected as the best subset, we also determined the best subset of exactly 2, 3, 4, and five biomarkers.

We also performed other approaches to investigate whether they gave similar results. These included backward elimination, forward selection, stepwise forward, stepwise backward, augmented backward elimination, and Bayesian projection variable selection (*Heinze et al., 2018*; *Piironen and Vehtari, 2017*; *Sauerbrei et al., 2020*).

## Checking model robustness by bootstrap resampling

To check the robustness (stability) of the selected 'best subset' model, we used a bootstrap procedure by resampling with replacement from the original data set (1000 times). For each bootstrap sample, we performed the 'best subset' approach (similar to step #2 above) to determine the best model based on the lowest AIC. From the 1000 samples we calculated *Heinze et al., 2018*:

i. Inclusion frequency for each of the ten biomarkers
ii. The root mean squared difference (RMSD) ratio of each regression coefficient. The root mean squared difference is computed between the 1000 estimates of the regression coefficient after the best subset selection and its value in the initial model (which includes all 10 biomarkers and is estimated on the original data).

$$RMSD(\beta_j) = \sqrt{\sum_b \frac{\left(\beta_{bootstrap,j}^{(b)} - \beta_{initial,j}\right)^2}{n_{bootstrap}}}$$

$\beta_{bootstrap,j}^{(b)}$ is the estimate of parameter j in bootstrap sample b

$\beta_{initial,j}$ is the estimate of parameter j based on the initial model

$n_{bootstrap}$ is the number of bootstrap samples (1000)

The RMSD ratio is the RMSD divided by the standard error of that coefficient in the initial model.

iii. Relative bias conditional on selection for each parameter

$$Relative\ bias = \left( \frac{\bar{\hat{\beta}}_{bootstrap}}{\hat{\beta}_{initial} \times BIF} - 1 \right) \times 100\%$$

$\bar{\beta}_{bootstrap}$ is the mean bootstrapped estimate of the parameter

$\beta_{initial}$ is the initial model estimate of the parameter

$BIF$ is the bootstrap inclusion frequency of the corresponding biomarker

iv. The selection frequencies for the finally selected model and the 20 most frequent selected models

v. The median and the 2.5th and 97.5th percentiles of the regression coefficient of each biomarker.

In ii, iii and v, the value of a parameter was set at zero if the marker was not selected.

## Appendix 4

### Additional descriptive analysis

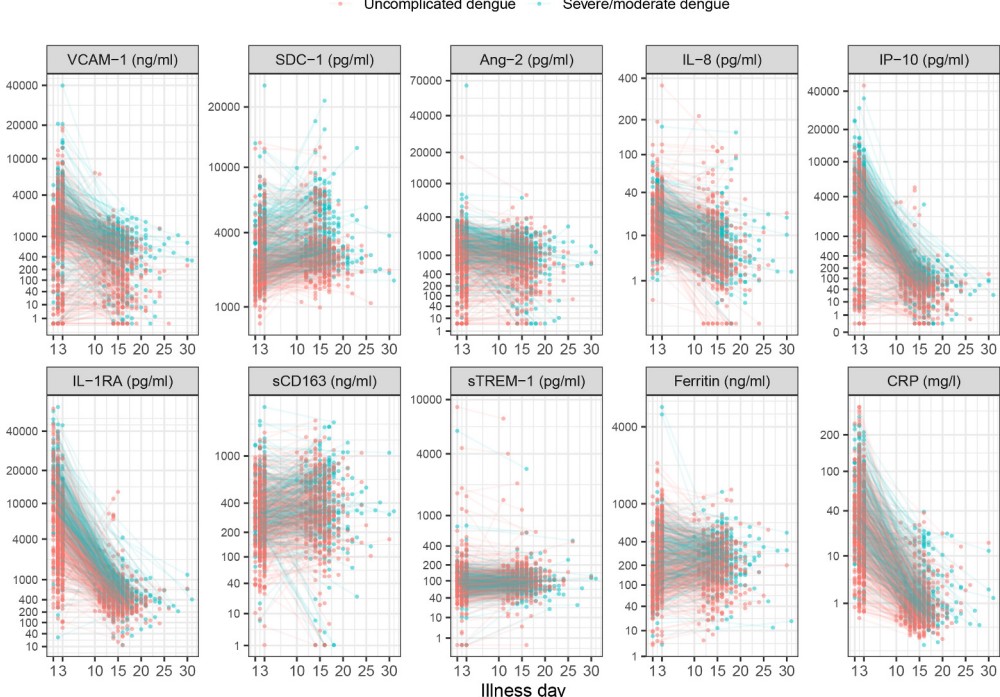

**Appendix 4—figure 1.** Biomarker levels by individual. Y-axes are transformed using the fourth root transformation.

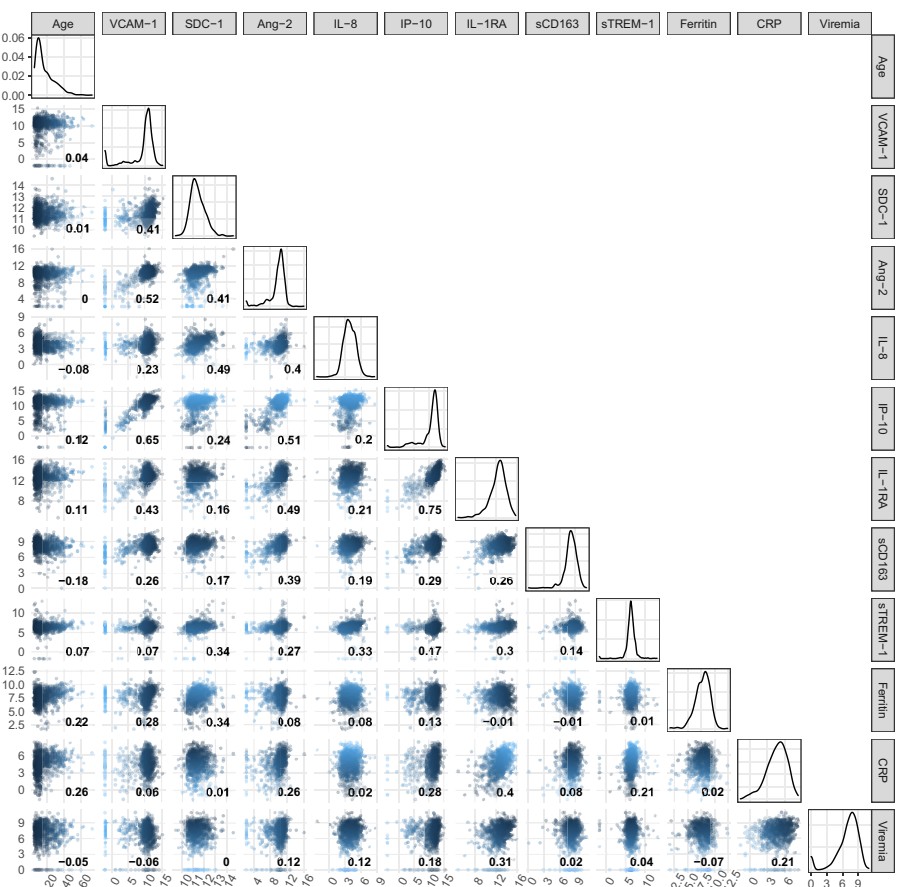

**Appendix 4—figure 2.** Pairwise correlation of biomarker levels at enrollment and age. Viremia levels are transformed using log-10. All other biomarker levels are transformed using log-2. The number inside each scatter plot represents the Spearman's rank correlation coefficient of the two variables at the corresponding column and row. When the column and row refer to the same variable, the corresponding scatter plot is replaced by a density plot to reflect the distribution of that biomarker. VCAM-1: vascular cell adhesion molecule-1; SDC-1: syndecan-1; Ang-2: angiopoietin-2; IL-8: interleukin-8; IP-10: interferon gamma-induced protein-10; IL-1RA: interleukin-1 receptor antagonist; sCD163: soluble cluster of differentiation 163; sTREM-1: soluble triggering receptor expressed on myeloid cells-1; CRP: C-reactive protein.

**Appendix 4—table 1.** Summary of clinical phenotype of the primary endpoint.

|  | All patients (N = 281) | Children (N = 127) | Adults (N = 154) |
| --- | --- | --- | --- |
| Severe dengue, n (%) | 38 (14) | 29 (23) | 9 (6) |
| - Severe plasma leakage | 33 (12) | 24 (19) | 9 (6) |
| +Dengue shock syndrome | 25 (9) | 18 (14) | 7 (5) |
| +Respiratory distress | 12 (4) | 9 (7) | 3 (2) |
| - Severe neurologic involvement | 3 (1) | 3 (2) | 0 (0) |
| - Severe bleeding | 2 (1) | 2 (2) | 0 (0) |
| - Severe other major organ failure | 1 (0) | 0 (0) | 1 (1) |
| - Severe hepatic involvement | 0 (0) | 0 (0) | 0 (0) |

*Continued on next page*

*Appendix 4—table 1 continued*

|  | All patients (N = 281) | Children (N = 127) | Adults (N = 154) |
|---|---|---|---|
| Moderate dengue, *n (%)* | 243 (86) | 98 (77) | 145 (94) |
| - Moderate plasma leakage | 159 (57) | 73 (57) | 86 (56) |
| - Moderate hepatic involvement | 102 (36) | 35 (28) | 67 (44) |
| - Moderate bleeding | 9 (3) | 3 (2) | 6 (4) |
| - Moderate other major organ involvement | 1 (0) | 0 (0) | 1 (1) |
| - Moderate neurologic involvement | 0 (0) | 0 (0) | 0 (0) |

**Appendix 4—table 2.** Summary of biomarkers' data.

|  | All patients | | Children | | Adults | |
|---|---|---|---|---|---|---|
|  | Uncomplicated dengue | Severe/moderate dengue | Uncomplicated dengue | Severe/moderate dengue | Uncomplicated dengue | Severe/moderate dengue |
| **At enrollment** | (N = 556) | (N = 281) | (N = 337) | (N = 127) | (N = 219) | (N = 154) |
| VCAM-1 (ng/ml) | 1404 (540, 2548) | 2027 (1122, 3577) | 1442 (447, 2546) | 2020 (1232, 3384) | 1356 (568, 2560) | 2092 (1060, 4202) |
| SDC-1 (pg/ml) | 2334 (1864, 3131) | 2997 (2230, 4201) | 2369 (1861, 3423) | 2846 (2164, 4173) | 2260 (1879, 2898) | 3122 (2278, 4211) |
| Ang-2 (pg/ml) | 1064 (550, 1584) | 1521 (899, 2318) | 1102 (584, 1563) | 1547 (967, 2318) | 944 (516, 1585) | 1516 (885, 2321) |
| IL-8 (pg/ml) | 12 (8, 22) | 17 (11, 28) | 15 (9, 26) | 16 (10, 27) | 10 (7, 15) | 19 (12, 29) |
| IP-10 (pg/ml) | 2502 (732, 4509) | 4092 (2436, 6441) | 2245 (458, 4531) | 3942 (2046, 6287) | 2793 (1370, 4495) | 4242 (2524, 6469) |
| IL-1RA (pg/ml) | 5237 (2603, 9082) | 9105 (5933, 14977) | 4491 (2318, 8977) | 9688 (6109, 16786) | 5721 (3479, 9703) | 8993 (5953, 12935) |
| sCD163 (ng/ml) | 278 (185, 447) | 322 (228, 503) | 326 (212, 481) | 386 (256, 603) | 226 (157, 374) | 291 (207, 410) |
| sTREM-1 (pg/ml) | 81 (59, 114) | 96 (69, 132) | 80 (58, 115) | 93 (67, 128) | 84 (60, 114) | 98 (73, 134) |
| Ferritin (ng/ml) | 233 (116, 406) | 261 (133, 433) | 177 (99, 324) | 224 (110, 402) | 303 (161, 510) | 278 (160, 448) |
| CRP (mg/l) | 25 (10, 54) | 34 (17, 72) | 18 (7, 41) | 24 (13, 58) | 38 (17, 65) | 45 (25, 80) |
| Viremia ($10^6$ copies/ml) | 15.8 (0.7, 148.5) | 79.2 (5.3, 582.7) | 21.8 (1.8, 167.0) | 105.4 (8.4, 646.0) | 9.8 (0.3, 115.5) | 56.2 (3.6, 496.0) |
| **At follow-up** | (N = 437) | (N = 231) | (N = 292) | (N = 112) | (N = 145) | (N = 119) |
| VCAM-1 (ng/ml) | 402 (102, 730) | 686 (344, 961) | 579 (182, 858) | 782 (402, 1078) | 173 (26, 388) | 622 (343, 835) |
| SDC-1 (pg/ml) | 2769 (2298, 3514) | 3417 (2815, 5495) | 2957 (2319, 4115) | 3122 (2748, 5507) | 2666 (2196, 3058) | 3745 (2971, 5495) |
| Ang-2 (pg/ml) | 953 (478, 1479) | 1155 (675, 1567) | 1163 (738, 1646) | 1352 (710, 1856) | 565 (302, 923) | 1044 (626, 1345) |
| IL-8 (pg/ml) | 4.9 (2.3, 12.4) | 5.7 (2.7, 10.4) | 6.8 (3.0, 15.1) | 5.9 (2.4, 10.5) | 2.7 (1.6, 4.8) | 5.5 (3.1, 10.4) |
| IP-10 (pg/ml) | 57 (24, 91) | 76 (47, 133) | 67 (33, 98) | 86 (38, 143) | 39 (22, 70) | 75 (48, 108) |
| IL-1RA (pg/ml) | 412 (279, 635) | 455 (328, 626) | 441 (323, 687) | 501 (352, 664) | 336 (210, 480) | 407 (308, 615) |
| sCD163 (ng/ml) | 337 (216, 553) | 412 (257, 661) | 340 (226, 562) | 456 (279, 680) | 328 (199, 523) | 386 (241, 589) |
| sTREM-1 (pg/ml) | 99 (73, 132) | 90 (68, 116) | 98 (72, 132) | 91 (67, 115) | 101 (73, 134) | 90 (70, 116) |
| Ferritin (ng/ml) | 202 (120, 309) | 273 (181, 382) | 177 (112, 263) | 209 (154, 311) | 267 (160, 404) | 322 (247, 436) |
| CRP* (mg/l) | 0.7 (0.3, 1.8) | 0.8 (0.4, 2.0) | 0.6 (0.3, 1.3) | 0.6 (0.3, 1.1) | 1.1 (0.5, 2.7) | 1.1 (0.5, 3.4) |

*The number of cases with available data for CRP at follow-up in the uncomplicated and severe/moderate dengue groups are 436 and 228 (all patients); 292 and 111 (children); and 218 and 152 (adults) respectively.

VCAM-1: vascular cell adhesion molecule-1; SDC-1: syndecan-1; Ang-2: angiopoietin-2; IL-8: interleukin-8; IP-10: interferon gamma-induced protein-10; IL-1RA: interleukin-1 receptor antagonist; sCD163: soluble cluster of differentiation 163; sTREM-1: soluble triggering receptor expressed on myeloid cells-1; CRP: C-reactive protein.

Summary statistics are median (1st and 3rd quartiles).

## Appendix 5

### Results for primary endpoint in DENV-1 and other serotypes

As DENV-1 is predominant in this study (42% of all cases), we performed a sensitivity analysis taking into account potential differences of the association between the biomarkers and the primary endpoint. For this analysis, we performed the single and global models similar to the main analysis, plus including the interaction between the biomarkers and serotype. Serotype was treated as a binary variable with values of DENV-1 and non-DENV-1 due to the limited number of outcomes (severe/moderate dengue). We also performed the analysis with the four serotypes (DENV-1, 2, 3 and 4) but the results were not certain (confidence intervals were wide) as the limited sample size (results are not presented).

Below are the results from the single models (*Appendix 5—figure 1*, *Appendix 5—table 1*) and global model (*Appendix 5—figure 2*, *Appendix 5—table 2*). The results are reported separately for DENV-1 and other serotypes, and for children and adults. Overall, there was no significant difference between DENV-1 and the others, in both the single and global models. These results suggest that the association between biomarkers and clinical outcome is similar in patients infected with different DENV serotypes.

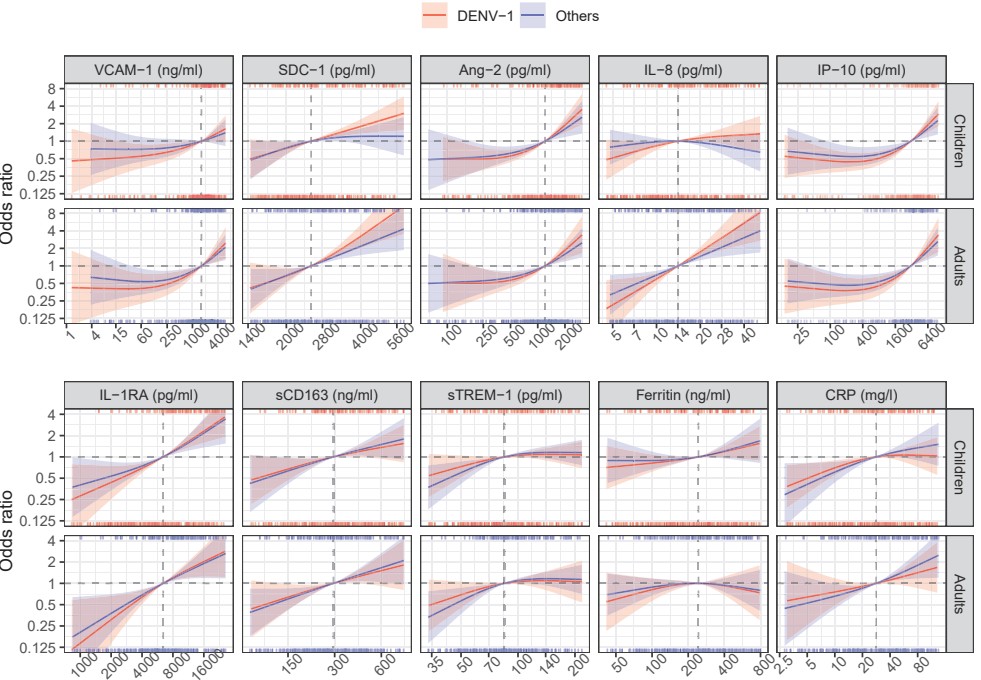

**Appendix 5—figure 1.** Results from single models for severe/moderate dengue with the interaction with serotype.

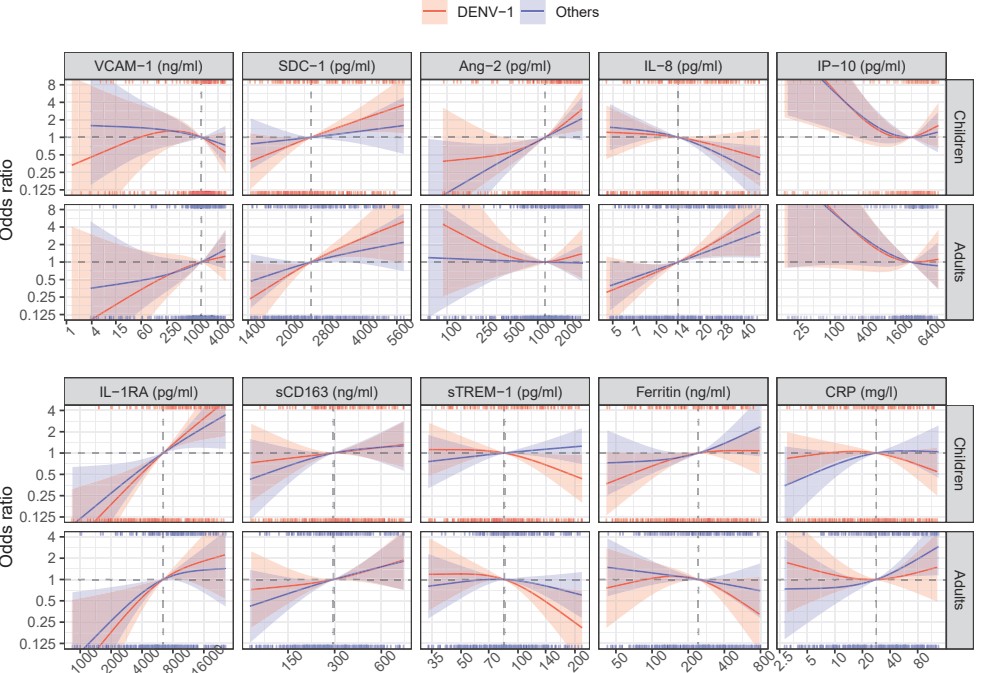

**Appendix 5—figure 2.** Results from global model for severe/moderate dengue with the interaction with serotype.

**Appendix 5—table 1.** Results from single models for severe/moderate dengue with the interaction with serotype.

| | Children | | Adults | | | | | |
| --- | --- | --- | --- | --- | --- | --- | --- | --- |
| | DENV-1 OR (95% CI) | Other serotypes OR (95% CI) | DENV-1 OR (95% CI) | Other serotypes OR (95% CI) | $P_{overall}$ | $P_1$ | $P_2$ | $P_3$ |
| VCAM-1 (ng/ml) | | | | | 0.008 | 0.822 | 0.729 | 0.565 |
| - 1636 vs 818 | 1.22 (1.02–1.46) | 1.14 (0.95–1.37) | 1.42 (1.13–1.77) | 1.32 (1.10–1.58) | | | | |
| - 3272 vs 1636 | 1.28 (0.99–1.66) | 1.18 (0.91–1.54) | 1.57 (1.15–2.14) | 1.45 (1.13–1.86) | | | | |
| SDC-1 (pg/ml) | | | | | <0.001 | 0.087 | 0.326 | 0.352 |
| - 2519 vs 1260 | 2.35 (0.89–6.25) | 2.56 (0.97–6.74) | 2.85 (0.80–10.12) | 3.10 (1.11–8.64) | | | | |
| - 5039 vs 2519 | 2.41 (1.47–3.96) | 1.22 (0.69–2.13) | 6.39 (2.81–14.53) | 3.23 (1.73–6.01) | | | | |
| Ang-2 (pg/ml) | | | | | <0.001 | 0.935 | 0.923 | 0.702 |
| - 1204 vs 602 | 1.67 (1.35–2.05) | 1.51 (1.21–1.88) | 1.64 (1.28–2.10) | 1.48 (1.20–1.83) | | | | |
| - 2409 vs 1204 | 2.46 (1.60–3.79) | 1.97 (1.25–3.11) | 2.40 (1.44–4.02) | 1.92 (1.28–2.89) | | | | |
| IL-8 (pg/ml) | | | | | <0.001 | <0.001 | <0.001 | 0.104 |
| - 14 vs 7 | 1.52 (1.00–2.31) | 1.12 (0.76–1.65) | 2.83 (1.57–5.12) | 2.09 (1.34–3.26) | | | | |
| - 28 vs 14 | 1.22 (0.90–1.65) | 0.84 (0.60–1.17) | 3.04 (1.88–4.94) | 2.10 (1.45–3.04) | | | | |
| IP-10 (pg/ml) | | | | | <0.001 | 0.975 | 0.950 | 0.681 |
| - 3093 vs 1546 | 1.54 (1.28–1.84) | 1.39 (1.13–1.70) | 1.64 (1.28–2.10) | 1.48 (1.20–1.83) | | | | |
| - 6186 vs 3093 | 1.84 (1.39–2.43) | 1.60 (1.17–2.20) | 2.00 (1.37–2.92) | 1.75 (1.27–2.40) | | | | |
| IL-1RA (pg/ml) | | | | | <0.001 | 0.577 | 0.280 | 0.805 |
| - 6434 vs 3217 | 1.68 (1.29–2.18) | 1.51 (1.20–1.90) | 1.96 (1.34–2.86) | 1.76 (1.29–2.38) | | | | |
| - 12868 vs 6434 | 1.87 (1.43–2.44) | 1.77 (1.28–2.46) | 1.73 (1.23–2.44) | 1.64 (1.18–2.29) | | | | |
| sCD163 (ng/ml) | | | | | 0.002 | 0.983 | 0.831 | 0.932 |
| - 295 vs 147 | 1.50 (1.03–2.20) | 1.60 (1.03–2.48) | 1.59 (1.02–2.47) | 1.69 (1.18–2.42) | | | | |
| - 589 vs 295 | 1.35 (0.94–1.94) | 1.48 (0.99–2.21) | 1.50 (0.92–2.43) | 1.64 (1.04–2.59) | | | | |
| sTREM-1 (pg/ml) | | | | | 0.146 | 0.979 | 0.998 | 0.597 |
| - 85 vs 42 | 1.55 (0.94–2.57) | 2.03 (1.22–3.39) | 1.68 (0.92–3.07) | 2.20 (1.21–4.02) | | | | |
| - 169 vs 85 | 1.08 (0.80–1.45) | 1.16 (0.87–1.54) | 1.08 (0.68–1.71) | 1.16 (0.83–1.60) | | | | |
| Ferritin (ng/ml) | | | | | 0.112 | 0.139 | 0.177 | 0.711 |
| - 243 vs 122 | 1.18 (0.97–1.44) | 1.12 (0.91–1.38) | 1.14 (0.88–1.49) | 1.08 (0.88–1.32) | | | | |
| - 487 vs 243 | 1.28 (0.96–1.71) | 1.32 (0.91–1.92) | 0.87 (0.55–1.38) | 0.90 (0.64–1.27) | | | | |
| CRP (mg/l) | | | | | <0.001 | 0.080 | 0.029 | 0.755 |
| - 28 vs 14 | 1.23 (1.06–1.43) | 1.36 (1.12–1.66) | 1.21 (0.94–1.57) | 1.34 (1.07–1.69) | | | | |
| - 56 vs 28 | 1.06 (0.85–1.33) | 1.24 (0.97–1.58) | 1.25 (0.93–1.68) | 1.46 (1.13–1.87) | | | | |

Odds ratios are estimated at age of 10 and 25 years, represented as children and adults respectively; $P_{overall}$ is derived from Wald test for the overall association of the biomarker with the endpoint; $P_1$ is from the test for the overall interaction of the biomarker; $P_2$ is from the test for the interaction between the biomarker and age; $P_3$ is from the test for the interaction between the biomarker and serotype.

**Appendix 5—table 2.** Results from global model for severe/moderate dengue with the interaction with serotype.

| | Children | | Adults | | | | | |
|---|---|---|---|---|---|---|---|---|
| | DENV-1 OR (95% CI) | Other serotypes OR (95% CI) | DENV-1 OR (95% CI) | Other serotypes OR (95% CI) | $P_{overall}$ | $P_1$ | $P_2$ | $P_3$ |
| VCAM-1 (ng/ml) | | | | | 0.449 | 0.258 | 0.248 | 0.327 |
| - 1636 vs 818 | 0.84 (0.62–1.13) | 0.88 (0.66–1.17) | 1.18 (0.81–1.72) | 1.24 (0.93–1.65) | | | | |
| - 3272 vs 1636 | 0.75 (0.50–1.12) | 0.85 (0.58–1.25) | 1.14 (0.70–1.85) | 1.29 (0.88–1.90) | | | | |
| SDC-1 (pg/ml) | | | | | 0.027 | 0.788 | 0.821 | 0.316 |
| - 2519 vs 1260 | 3.21 (0.79–12.94) | 1.38 (0.38–4.94) | 5.98 (1.00–35.72) | 2.57 (0.66–10.01) | | | | |
| - 5039 vs 2519 | 2.82 (1.21–6.57) | 1.45 (0.62–3.40) | 3.70 (1.08–12.72) | 1.90 (0.80–4.52) | | | | |
| Ang-2 (pg/ml) | | | | | 0.067 | 0.102 | 0.043 | 0.472 |
| - 1204 vs 602 | 1.65 (1.10–2.47) | 1.79 (1.18–2.72) | 0.89 (0.55–1.44) | 0.97 (0.67–1.40) | | | | |
| - 2409 vs 1204 | 2.22 (1.21–4.05) | 1.73 (0.94–3.17) | 1.25 (0.63–2.48) | 0.97 (0.58–1.62) | | | | |
| IL-8 (pg/ml) | | | | | <0.001 | <0.001 | <0.001 | 0.591 |
| - 14 vs 7 | 0.86 (0.48–1.51) | 0.73 (0.44–1.24) | 2.14 (1.01–4.53) | 1.84 (1.05–3.20) | | | | |
| - 28 vs 14 | 0.68 (0.41–1.13) | 0.49 (0.29–0.82) | 2.60 (1.27–5.32) | 1.88 (1.20–2.96) | | | | |
| IP-10 (pg/ml) | | | | | 0.068 | 0.875 | 0.715 | 0.888 |
| - 3093 vs 1546 | 0.98 (0.69–1.40) | 0.91 (0.65–1.26) | 0.86 (0.53–1.40) | 0.80 (0.53–1.19) | | | | |
| - 6186 vs 3093 | 1.27 (0.77–2.10) | 1.10 (0.70–1.74) | 1.04 (0.54–2.00) | 0.90 (0.53–1.53) | | | | |
| IL-1RA (pg/ml) | | | | | <0.001 | 0.333 | 0.230 | 0.711 |
| - 6434 vs 3217 | 2.63 (1.65–4.20) | 2.11 (1.32–3.39) | 2.50 (1.28–4.88) | 2.01 (1.15–3.51) | | | | |
| - 12868 vs 6434 | 2.38 (1.45–3.91) | 1.90 (1.19–3.04) | 1.65 (0.88–3.09) | 1.32 (0.78–2.23) | | | | |
| sCD163 (ng/ml) | | | | | 0.340 | 0.661 | 0.455 | 0.769 |
| - 295 vs 147 | 1.20 (0.65–2.20) | 1.54 (0.83–2.87) | 1.24 (0.68–2.27) | 1.61 (0.92–2.80) | | | | |
| - 589 vs 295 | 1.20 (0.77–1.87) | 1.20 (0.75–1.94) | 1.49 (0.84–2.67) | 1.50 (0.85–2.61) | | | | |
| sTREM-1 (pg/ml) | | | | | 0.441 | 0.289 | 0.306 | 0.071 |
| - 85 vs 42 | 0.90 (0.48–1.67) | 1.23 (0.66–2.28) | 0.84 (0.36–1.94) | 1.15 (0.54–2.42) | | | | |
| - 169 vs 85 | 0.56 (0.33–0.96) | 1.18 (0.79–1.77) | 0.34 (0.15–0.76) | 0.71 (0.43–1.19) | | | | |
| Ferritin (ng/ml) | | | | | 0.067 | 0.033 | 0.013 | 0.331 |
| - 243 vs 122 | 1.36 (1.00–1.85) | 1.25 (0.92–1.69) | 0.92 (0.62–1.36) | 0.84 (0.63–1.13) | | | | |
| - 487 vs 243 | 1.08 (0.72–1.63) | 1.59 (0.97–2.61) | 0.55 (0.29–1.05) | 0.81 (0.51–1.29) | | | | |
| CRP (mg/l) | | | | | 0.156 | 0.103 | 0.241 | 0.136 |
| - 28 vs 14 | 0.95 (0.78–1.16) | 1.25 (0.98–1.60) | 0.94 (0.67–1.32) | 1.24 (0.90–1.70) | | | | |
| - 56 vs 28 | 0.80 (0.60–1.07) | 1.07 (0.80–1.44) | 1.13 (0.75–1.70) | 1.51 (1.09–2.09) | | | | |

Odds ratios are estimated at age of 10 and 25 years, represented as children and adults respectively; $P_{overall}$ is derived from Wald test for the overall association of the biomarker with the endpoint; $P_1$ is from the test for the overall interaction of the biomarker; $P_2$ is from the test for the interaction between the biomarker and age; $P_3$ is from the test for the interaction between the biomarker and serotype.

## Appendix 6

### Results for secondary endpoints

In the single models, higher levels of the biomarkers generally increased the risk of developing severe dengue (SD), however, as the number of events was small, the confidence intervals (CIs) were wide and the association was not certain (*Appendix 6—figure 1*, *Appendix 6—table 1*). For severe dengue or dengue with warning signs (SD/DWWS) and hospitalization endpoints, the associations were similar to the primary endpoint, apart for sCD163, sTREM-1, and CRP (*Appendix 6—figure 2*, *Appendix 6—figure 3*, *Appendix 6—table 2*, *Appendix 6—table 3*) – these biomarkers did not show an association with the endpoints. Moreover, the odds ratios (ORs) of SD/DWWS and hospitalization were generally lower than of severe or moderate dengue (S/MD) for every 2-fold difference in biomarker levels.

The difference between the global and single models in the analysis of secondary endpoints was similar to in the primary endpoint. The most stable biomarkers were SDC-1 and IL-1RA, while IP-10 markedly changed the trend of the association with the endpoints; others showed a weaker association.

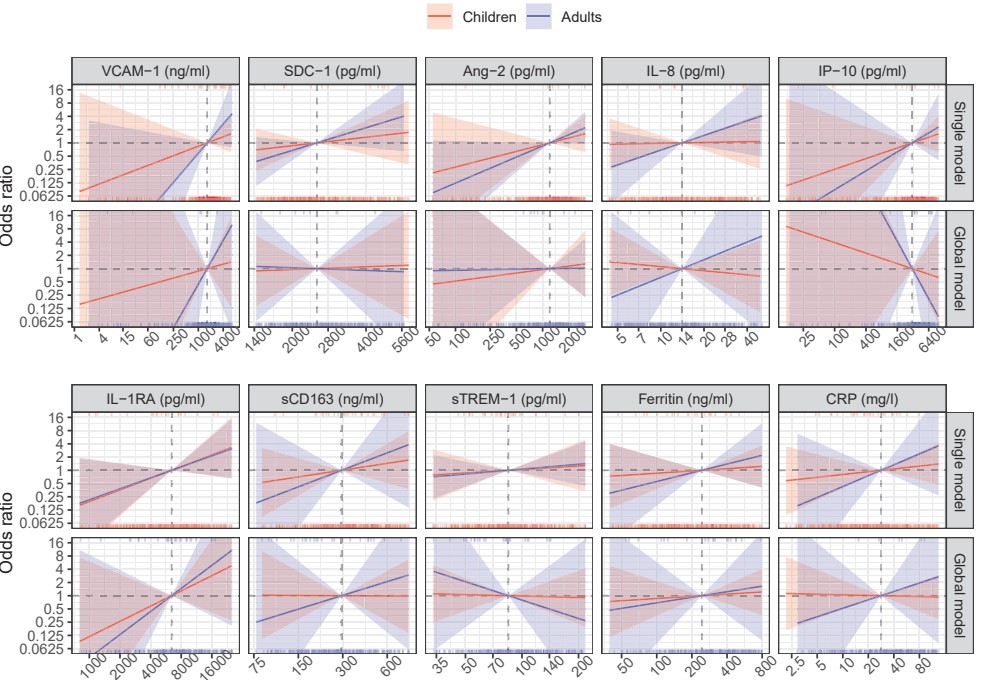

**Appendix 6—figure 1.** Results from models for severe dengue endpoint.

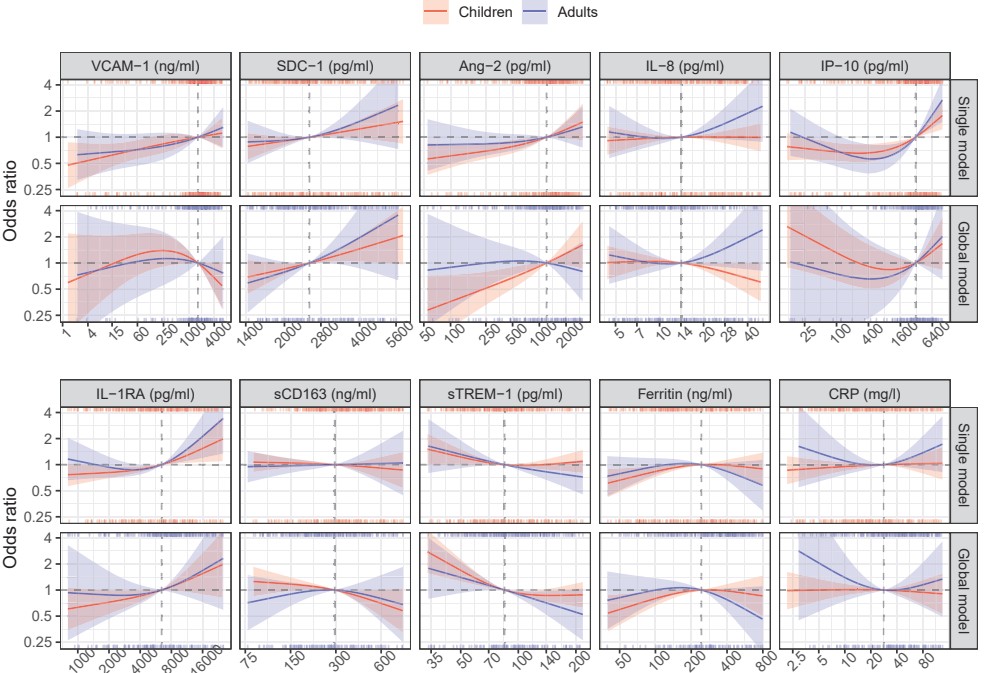

**Appendix 6—figure 2.** Results from models for severe dengue or dengue with warning signs endpoint.

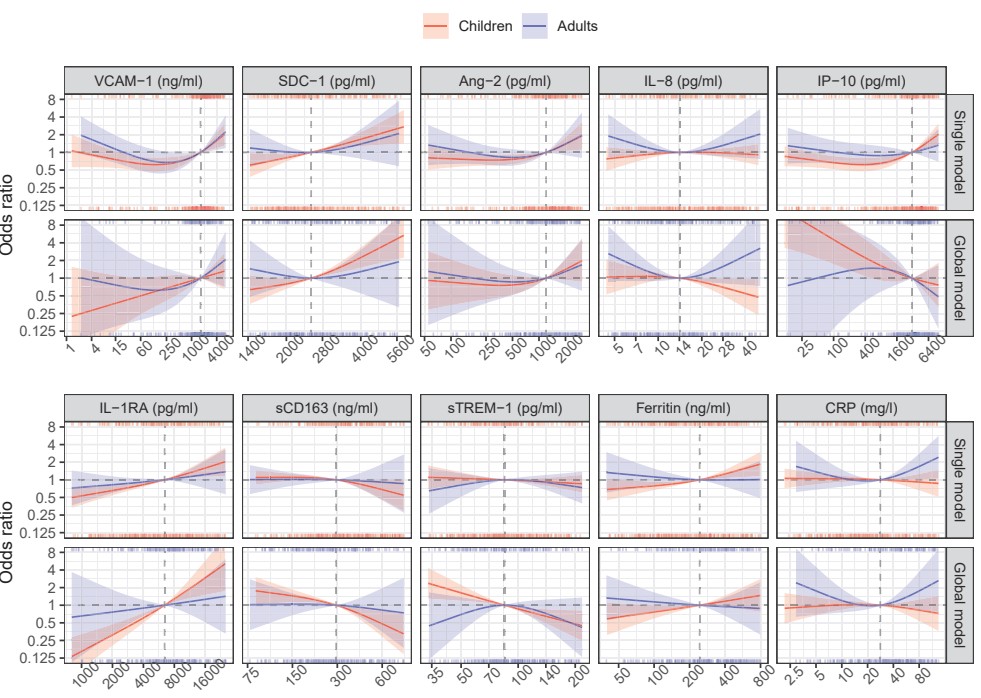

**Appendix 6—figure 3.** Results from models for hospitalization endpoint.

**Appendix 6—table 1.** Results from models for severe dengue endpoint.

| | Single models | | | | Global model | | | |
|---|---|---|---|---|---|---|---|---|
| | Children OR (95% CI) | Adults OR (95% CI) | $P_{overall}$ | $P_{interaction}$ | Children OR (95% CI) | Adults OR (95% CI) | $P_{overall}$ | $P_{interaction}$ |
| VCAM-1 (ng/ml) | 1.28 (0.77–2.12) | 2.13 (0.88–5.15) | 0.236 | 0.202 | 1.20 (0.35–4.12) | 3.13 (0.19–50.77) | 0.723 | 0.491 |
| SDC-1 (pg/ml) | 1.55 (0.41–5.96) | 3.26 (0.69–15.52) | 0.307 | 0.438 | 1.16 (0.12–11.07) | 0.87 (0.03–27.93) | 0.987 | 0.884 |
| Ang-2 (pg/ml) | 1.42 (0.70–2.88) | 1.79 (0.97–3.31) | 0.148 | 0.580 | 1.20 (0.33–4.31) | 1.02 (0.33–3.19) | 0.961 | 0.830 |
| IL-8 (pg/ml) | 1.04 (0.47–2.33) | 2.15 (0.67–6.88) | 0.420 | 0.338 | 0.81 (0.28–2.31) | 2.51 (0.16–38.43) | 0.741 | 0.446 |
| IP-10 (pg/ml) | 1.32 (0.75–2.33) | 1.64 (0.58–4.62) | 0.443 | 0.708 | 0.76 (0.17–3.30) | 0.22 (0.01–7.12) | 0.693 | 0.473 |
| IL-1RA (pg/ml) | 1.84 (0.81–4.16) | 1.78 (0.80–3.94) | 0.098 | 0.956 | 2.22 (0.51–9.64) | 3.35 (0.45–24.87) | 0.386 | 0.696 |
| sCD163 (ng/ml) | 1.43 (0.52–3.94) | 2.43 (0.27–21.70) | 0.617 | 0.645 | 0.98 (0.27–3.59) | 2.06 (0.15–27.47) | 0.855 | 0.590 |
| sTREM-1 (pg/ml) | 1.20 (0.44–3.28) | 1.29 (0.55–3.05) | 0.793 | 0.914 | 0.93 (0.30–2.88) | 0.38 (0.02–8.76) | 0.827 | 0.541 |
| Ferritin (ng/ml) | 1.13 (0.59–2.16) | 1.58 (0.59–4.23) | 0.659 | 0.501 | 1.12 (0.55–2.29) | 1.34 (0.25–7.02) | 0.926 | 0.819 |
| CRP (mg/l) | 1.16 (0.71–1.88) | 1.77 (0.56–5.62) | 0.620 | 0.418 | 0.97 (0.58–1.63) | 1.55 (0.33–7.32) | 0.773 | 0.500 |

Odds ratios (95% confidence intervals) are calculated for each 2-fold increase of the biomarkers and are estimated at age of 10 and 25 years, represented as children and adults respectively; $P_{overall}$ is derived from Wald test for the overall association of the biomarker with the endpoint; $P_{interaction}$ is from the test for the interaction between the biomarker and age.

**Appendix 6—table 2.** Results from models for severe dengue or dengue with warning signs endpoint.

| | Single models | | | | Global model | | | |
|---|---|---|---|---|---|---|---|---|
| | Children OR (95% CI) | Adults OR (95% CI) | $P_{overall}$ | $P_{interaction}$ | Children OR (95% CI) | Adults OR (95% CI) | $P_{overall}$ | $P_{interaction}$ |
| VCAM-1 (ng/ml) | | | 0.025 | 0.374 | | | 0.469 | 0.763 |
| - 1636 vs 818 | 1.06 (0.93–1.22) | 1.11 (0.92–1.33) | | | 0.82 (0.66–1.02) | 0.92 (0.66–1.30) | | |
| - 3272 vs 1636 | 1.06 (0.87–1.29) | 1.13 (0.88–1.46) | | | 0.74 (0.55–1.01) | 0.88 (0.56–1.40) | | |
| SDC-1 (pg/ml) | | | 0.032 | 0.363 | | | 0.116 | 0.773 |
| - 2519 vs 1260 | 1.34 (0.86–2.09) | 1.15 (0.56–2.33) | | | 1.57 (0.89–2.75) | 1.89 (0.71–5.03) | | |
| - 5039 vs 2519 | 1.40 (0.90–2.17) | 2.00 (0.79–5.08) | | | 1.78 (1.00–3.18) | 2.88 (0.71–11.69) | | |
| Ang-2 (pg/ml) | | | 0.008 | 0.637 | | | 0.009 | 0.011 |
| - 1204 vs 602 | 1.23 (1.05–1.44) | 1.12 (0.93–1.36) | | | 1.37 (1.07–1.75) | 0.95 (0.69–1.30) | | |
| - 2409 vs 1204 | 1.34 (0.95–1.88) | 1.22 (0.82–1.80) | | | 1.42 (0.91–2.21) | 0.85 (0.48–1.51) | | |
| IL-8 (pg/ml) | | | 0.040 | 0.020 | | | 0.053 | 0.030 |
| - 14 vs 7 | 1.05 (0.87–1.27) | 0.97 (0.67–1.41) | | | 0.96 (0.77–1.21) | 0.94 (0.62–1.42) | | |
| - 28 vs 14 | 1.01 (0.85–1.19) | 1.45 (0.94–2.22) | | | 0.78 (0.62–0.99) | 1.48 (0.89–2.44) | | |
| IP-10 (pg/ml) | | | <0.001 | 0.176 | | | 0.059 | 0.537 |
| - 3093 vs 1546 | 1.26 (1.09–1.44) | 1.44 (1.16–1.80) | | | 1.16 (0.87–1.55) | 1.30 (0.81–2.09) | | |
| - 6186 vs 3093 | 1.39 (1.13–1.71) | 1.75 (1.26–2.44) | | | 1.33 (0.89–1.99) | 1.49 (0.78–2.87) | | |
| IL-1RA (pg/ml) | | | 0.005 | 0.381 | | | 0.425 | 0.955 |
| - 6434 vs 3217 | 1.17 (1.06–1.30) | 1.15 (0.97–1.36) | | | 1.24 (1.01–1.52) | 1.14 (0.80–1.63) | | |
| - 12868 vs 6434 | 1.37 (1.06–1.77) | 1.70 (1.13–2.55) | | | 1.38 (0.93–2.05) | 1.45 (0.80–2.64) | | |
| sCD163 (ng/ml) | | | 0.854 | 0.719 | | | 0.193 | 0.419 |
| - 295 vs 147 | 0.96 (0.84–1.08) | 1.03 (0.85–1.25) | | | 0.85 (0.71–1.03) | 1.12 (0.81–1.55) | | |
| - 589 vs 295 | 0.91 (0.69–1.22) | 1.03 (0.61–1.73) | | | 0.71 (0.50–1.00) | 0.80 (0.44–1.47) | | |
| sTREM-1 (pg/ml) | | | 0.221 | 0.472 | | | 0.002 | 0.132 |
| - 85 vs 42 | 0.75 (0.56–1.00) | 0.69 (0.41–1.16) | | | 0.48 (0.32–0.73) | 0.64 (0.36–1.16) | | |
| - 169 vs 85 | 1.04 (0.84–1.28) | 0.78 (0.57–1.07) | | | 0.87 (0.69–1.10) | 0.62 (0.39–1.00) | | |
| Ferritin (ng/ml) | | | 0.034 | 0.258 | | | 0.024 | 0.075 |
| - 243 vs 122 | 1.13 (1.02–1.26) | 1.01 (0.85–1.19) | | | 1.17 (1.01–1.35) | 0.96 (0.76–1.23) | | |
| - 487 vs 243 | 0.96 (0.77–1.20) | 0.76 (0.53–1.08) | | | 0.94 (0.71–1.25) | 0.67 (0.42–1.07) | | |
| CRP (mg/l) | | | 0.747 | 0.622 | | | 0.662 | 0.448 |
| - 28 vs 14 | 1.03 (0.96–1.12) | 0.97 (0.79–1.20) | | | 0.99 (0.89–1.10) | 0.84 (0.59–1.20) | | |
| - 56 vs 28 | 1.02 (0.87–1.19) | 1.19 (0.92–1.54) | | | 0.96 (0.79–1.17) | 1.06 (0.75–1.49) | | |

Odds ratios are estimated at age of 10 and 25 years, represented as children and adults respectively; $P_{overall}$ is derived from Wald test for the overall association of the biomarker with the endpoint; $P_{interaction}$ is from the test for the interaction between the biomarker and age.

**Appendix 6—table 3.** Results from models for hospitalization endpoint.

| | Single models | | | | Global model | | | |
|---|---|---|---|---|---|---|---|---|
| | Children OR (95% CI) | Adults OR (95% CI) | $P_{overall}$ | $P_{interaction}$ | Children OR (95% CI) | Adults OR (95% CI) | $P_{overall}$ | $P_{interaction}$ |
| VCAM-1 (ng/ml) | | | <0.001 | 0.009 | | | 0.092 | 0.137 |
| - 1636 vs 818 | 1.28 (1.11–1.49) | 1.28 (1.03–1.60) | | | 1.16 (0.90–1.48) | 1.28 (0.87–1.90) | | |
| - 3272 vs 1636 | 1.42 (1.14–1.76) | 1.46 (1.08–1.97) | | | 1.16 (0.81–1.64) | 1.41 (0.84–2.37) | | |
| SDC-1 (pg/ml) | | | <0.001 | 0.187 | | | 0.006 | 0.406 |
| - 2519 vs 1260 | 1.82 (1.01–3.28) | 0.79 (0.31–2.05) | | | 1.70 (0.84–3.41) | 0.62 (0.16–2.49) | | |
| - 5039 vs 2519 | 2.22 (1.36–3.63) | 1.81 (0.65–5.07) | | | 3.70 (1.90–7.22) | 1.66 (0.39–7.07) | | |
| Ang-2 (pg/ml) | | | 0.012 | 0.337 | | | 0.497 | 0.789 |
| - 1204 vs 602 | 1.27 (1.07–1.52) | 1.21 (0.91–1.62) | | | 1.27 (0.92–1.77) | 1.16 (0.76–1.77) | | |
| - 2409 vs 1204 | 1.58 (1.08–2.32) | 1.61 (0.86–3.04) | | | 1.63 (0.90–2.93) | 1.45 (0.70–3.01) | | |
| IL-8 (pg/ml) | | | 0.007 | 0.002 | | | 0.024 | 0.021 |
| - 14 vs 7 | 1.14 (0.90–1.44) | 0.73 (0.47–1.15) | | | 0.94 (0.66–1.33) | 0.63 (0.36–1.13) | | |
| - 28 vs 14 | 0.98 (0.81–1.18) | 1.32 (0.85–2.05) | | | 0.70 (0.50–0.97) | 1.59 (0.81–3.12) | | |
| IP-10 (pg/ml) | | | 0.002 | 0.242 | | | 0.005 | 0.212 |
| - 3093 vs 1546 | 1.32 (1.13–1.54) | 1.10 (0.86–1.40) | | | 0.80 (0.56–1.14) | 0.77 (0.45–1.30) | | |
| - 6186 vs 3093 | 1.50 (1.18–1.90) | 1.17 (0.81–1.70) | | | 0.84 (0.51–1.40) | 0.66 (0.32–1.35) | | |
| IL-1RA (pg/ml) | | | <0.001 | 0.685 | | | <0.001 | 0.389 |
| - 6434 vs 3217 | 1.31 (1.17–1.46) | 1.13 (0.94–1.36) | | | 2.05 (1.57–2.66) | 1.18 (0.73–1.89) | | |
| - 12868 vs 6434 | 1.41 (1.11–1.80) | 1.17 (0.79–1.72) | | | 2.25 (1.39–3.64) | 1.19 (0.62–2.28) | | |
| sCD163 (ng/ml) | | | 0.208 | 0.722 | | | 0.007 | 0.419 |
| - 295 vs 147 | 0.90 (0.79–1.04) | 0.98 (0.76–1.27) | | | 0.69 (0.53–0.89) | 0.96 (0.62–1.50) | | |
| - 589 vs 295 | 0.69 (0.47–1.00) | 0.91 (0.46–1.82) | | | 0.49 (0.30–0.80) | 0.83 (0.36–1.91) | | |
| sTREM-1 (pg/ml) | | | 0.635 | 0.371 | | | 0.011 | 0.053 |
| - 85 vs 42 | 0.93 (0.67–1.29) | 1.35 (0.71–2.58) | | | 0.53 (0.34–0.81) | 1.74 (0.67–4.52) | | |
| - 169 vs 85 | 0.89 (0.71–1.13) | 0.83 (0.54–1.28) | | | 0.55 (0.36–0.83) | 0.57 (0.26–1.29) | | |
| Ferritin (ng/ml) | | | <0.001 | 0.011 | | | 0.129 | 0.117 |
| - 243 vs 122 | 1.22 (1.08–1.38) | 0.92 (0.72–1.18) | | | 1.23 (1.02–1.49) | 0.91 (0.68–1.21) | | |
| - 487 vs 243 | 1.40 (1.10–1.79) | 0.99 (0.68–1.46) | | | 1.25 (0.89–1.75) | 0.93 (0.54–1.59) | | |
| CRP (mg/l) | | | 0.379 | 0.190 | | | 0.139 | 0.053 |
| - 28 vs 14 | 0.97 (0.89–1.06) | 1.01 (0.83–1.24) | | | 0.97 (0.86–1.11) | 0.95 (0.71–1.27) | | |
| - 56 vs 28 | 0.95 (0.78–1.15) | 1.35 (1.01–1.81) | | | 0.89 (0.69–1.15) | 1.37 (0.88–2.13) | | |

Odds ratios are estimated at age of 10 and 25 years, represented as children and adults respectively; $P_{overall}$ is derived from Wald test for the overall association of the biomarker with the endpoint; $P_{interaction}$ is from the test for the interaction between the biomarker and age.

# Appendix 7

## Results from bootstrap resampling to check model robustness

*Appendix 7—table 1*, *Appendix 7—table 2*, *Appendix 7—table 3*, *Appendix 7—table 4* show the robustness (stability) of the selected models. The best subset models for children and adults based on the original data ranked first (*Appendix 7—table 1*, *Appendix 7—table 3*), but they were selected in only 13.4% and 7.9% of the resamples, indicating the instability of these models. The almost equal inclusion frequencies of the models ranked next suggest that there are many competing models of the selected one. Variable selection also added to uncertainty about the regression coefficients of the parameters, which is evidenced by the RMSD ratio of more than one in most of the biomarkers, except for sTREM-1 (0.92) and CRP (0.88) in children (*Appendix 7—table 2* and *Appendix 7—table 4*). Regarding relative conditional bias, which quantifies expected bias induced by variable selection of a parameter when it is selected, it is quite small for the first 3–4 selected parameters (IL-1RA, Ang-2, IL-8, and ferritin for children, and SDC-1, IL-8, and ferritin for adults), all of which have bootstrap inclusion percentages greater than 90%. This bias is much higher in the parameters for which selection is less certain. The bootstrap median and percentiles of the regression coefficients of the biomarkers reflect the variability of the coefficients over the different models selected in the bootstrap samples. The coefficients of the selected parameters from the initial estimates and the bootstrap median were very similar, suggesting no selection bias in the selected model.

**Appendix 7—table 1.** Model selection frequencies for children.

| Model | Included variables | | | | | | | | | | Count | Percent |
|---|---|---|---|---|---|---|---|---|---|---|---|---|
| | VCAM-1 | SDC-1 | Ang-2 | IL-8 | IP-10 | IL-1RA | sCD163 | sTREM-1 | Ferritin | CRP | | |
| 1 | | + | + | + | + | + | | | + | | 134 | 13.4 |
| 2 | | + | + | + | + | + | + | | + | | 100 | 10.0 |
| 3 | | | + | + | + | + | + | | + | | 55 | 5.5 |
| 4 | | | + | + | + | + | | | + | | 54 | 5.4 |
| 5 | + | + | + | + | | + | + | | + | | 48 | 4.8 |
| 6 | + | + | + | + | + | + | | | + | | 47 | 4.7 |
| 7 | + | + | + | + | + | + | + | | + | | 46 | 4.6 |
| 8 | + | + | + | + | | + | | | + | | 40 | 4.0 |
| 9 | | + | + | + | + | + | | | + | + | 39 | 3.9 |
| 10 | | + | + | + | + | + | + | + | + | | 36 | 3.6 |
| 11 | | + | + | + | + | + | + | | + | + | 28 | 2.8 |
| 12 | | + | + | + | | + | | + | + | | 23 | 2.3 |
| 13 | + | | + | + | + | + | | | + | | 23 | 2.3 |
| 14 | + | | + | + | | + | | | + | | 17 | 1.7 |
| 15 | | + | + | + | + | + | | + | + | + | 15 | 1.5 |
| 16 | + | + | + | + | | + | | | + | + | 14 | 1.4 |
| 17 | | | + | + | + | + | | | + | + | 13 | 1.3 |
| 18 | + | | + | + | + | + | + | | + | | 12 | 1.2 |
| 19 | + | + | + | + | + | + | | + | + | | 12 | 1.2 |
| 20 | | | + | + | + | + | + | | + | + | 11 | 1.1 |

Selected model is ranked first (bold face).

**Appendix 7—table 2.** Model stability for children.

| Predictors | Initial model Estimate | Standard error | Selected model Estimate | Standard error | Bootstrap inclusion frequency (%) | RMSD ratio | Relative conditional bias (%) | Bootstrap median | Bootstrap 2.5th percentile | Bootstrap 97.5th percentile |
|---|---|---|---|---|---|---|---|---|---|---|
| (Intercept) | −20.6801 | 3.0499 | −19.1331 | 2.8649 | 100.0 | 1.1330 | −1.4929 | −20.3233 | −26.6990 | −14.0696 |
| IL-1RA | 0.7604 | 0.1443 | 0.7885 | 0.1386 | 100.0 | 1.2352 | 1.6338 | 0.7651 | 0.4408 | 1.1236 |
| Ang-2 | 0.5203 | 0.1542 | 0.5371 | 0.1457 | 98.2 | 1.1710 | 7.6528 | 0.5416 | 0.2268 | 0.8804 |
| IL-8 | −0.4250 | 0.1338 | −0.4290 | 0.1307 | 97.2 | 1.0597 | 4.5484 | −0.4341 | −0.6995 | 0 |
| Ferritin | 0.2951 | 0.1044 | 0.2923 | 0.1003 | 93.5 | 1.2233 | 11.1579 | 0.3118 | 0 | 0.5358 |
| IP-10 | −0.2145 | 0.1079 | −0.2598 | 0.0925 | 75.9 | 1.3272 | 32.1767 | −0.2398 | −0.4676 | 0 |
| SDC-1 | 0.5079 | 0.2487 | 0.4398 | 0.2302 | 73.5 | 1.2692 | 18.5555 | 0.4923 | 0 | 1.0035 |
| sCD163 | 0.1778 | 0.1433 | | | 45.8 | 1.1935 | 69.8797 | 0 | 0 | 0.4926 |
| VCAM-1 | −0.0500 | 0.0541 | | | 35.2 | 1.1525 | 125.6348 | 0 | −0.1763 | 0 |
| sTREM-1 | −0.0689 | 0.1375 | | | 20.0 | 0.9176 | 159.0696 | 0 | −0.3584 | 0.2162 |
| CRP | 0.0438 | 0.0712 | | | 19.2 | 0.8804 | 173.2343 | 0 | 0 | 0.1754 |

RMSD: root mean squared difference.

**Appendix 7—table 3.** Model selection frequencies for adults.

| Model | VCAM-1 | SDC-1 | Ang-2 | IL-8 | IP-10* | IL-1RA | sCD163 | sTREM-1 | Ferritin | CRP | Count | Percent |
|---|---|---|---|---|---|---|---|---|---|---|---|---|
| 1 | | + | | + | + | + | + | + | + | | 79 | 7.9 |
| 2 | + | + | | + | + | + | + | + | + | | 55 | 5.5 |
| 3 | + | + | | + | + | + | | + | + | | 36 | 3.6 |
| 4 | | + | | + | | + | + | + | + | | 33 | 3.3 |
| 5 | | + | | + | | | | + | + | + | 30 | 3.0 |
| 6 | | + | + | + | + | + | + | + | + | | 29 | 2.9 |
| 7 | | + | | + | | | + | + | + | + | 26 | 2.6 |
| 8 | | + | | + | | + | | | + | | 25 | 2.5 |
| 9 | | + | | + | | + | | + | + | + | 24 | 2.4 |
| 10 | | + | | + | + | | + | + | + | + | 20 | 2.0 |
| 11 | + | + | | + | | + | + | + | + | | 20 | 2.0 |
| 12 | + | + | | + | + | + | + | + | + | + | 20 | 2.0 |
| 13 | + | + | | + | + | + | + | | + | | 19 | 1.9 |
| 14 | | + | | + | + | | + | + | + | | 17 | 1.7 |
| 15 | | + | | + | | | | + | | + | 16 | 1.6 |
| 16 | | + | | + | | | + | | + | | 16 | 1.6 |
| 17 | | + | | + | | | + | + | + | | 16 | 1.6 |
| 18 | | + | | + | + | | + | | + | | 16 | 1.6 |
| 19 | | + | | + | + | + | | + | + | | 16 | 1.6 |
| 20 | | + | | + | | + | + | + | + | | 15 | 1.5 |

*Variable is kept as non-linear effect using natural cubic splines with three knots.

Selected model is ranked first (bold face).

**Appendix 7—table 4.** Model stability for adults.

| Predictors | Initial model Estimate | Standard error | Selected model Estimate | Standard error | Bootstrap inclusion frequency (%) | RMSD ratio | Relative conditional bias (%) | Bootstrap median | Bootstrap 2.5th percentile | Bootstrap 97.5th percentile |
|---|---|---|---|---|---|---|---|---|---|---|
| (Intercept) | −16.6085 | 3.2632 | −16.6780 | 3.2475 | 100.0 | 1.3263 | 2.0586 | −16.7262 | −26.4755 | −9.8095 |
| SDC-1 | 1.1524 | 0.2708 | 1.1740 | 0.2607 | 99.2 | 1.2745 | 3.0018 | 1.1767 | 0.5450 | 1.8615 |
| IL-8 | 0.5473 | 0.1427 | 0.5544 | 0.1411 | 98.9 | 1.2973 | 5.8880 | 0.5739 | 0.2490 | 0.9520 |
| Ferritin | −0.2785 | 0.0916 | −0.2682 | 0.0883 | 94.6 | 1.2479 | 7.9330 | −0.2845 | −0.5012 | 0 |
| sTREM-1 | −0.2961 | 0.1499 | −0.2864 | 0.1492 | 66.5 | 1.4191 | 28.1687 | −0.2807 | −0.6448 | 0 |
| IL-1RA | 0.2582 | 0.1583 | 0.2557 | 0.1427 | 62.3 | 1.4788 | 54.9829 | 0.2494 | 0 | 0.7255 |
| IP-10 (ns1) * | −1.4427 | 1.0592 | −0.8269 | 0.6118 | 59.8 | 1.6064 | 36.8995 | −0.2108 | −5.2628 | 0.7003 |
| IP-10 (ns2) * | −0.1027 | 0.5763 | 0.1139 | 0.4976 | 59.8 | 1.2060 | 43.1473 | 0 | −1.7021 | 1.2589 |
| sCD163 | 0.2056 | 0.1287 | 0.2351 | 0.1253 | 59.1 | 1.3333 | 51.3932 | 0.2148 | 0 | 0.4989 |
| CRP | 0.0863 | 0.0990 | | | 36.3 | 1.1203 | 129.8186 | 0 | 0 | 0.3048 |
| VCAM-1 | 0.0660 | 0.0685 | | | 35.4 | 1.3808 | 98.3111 | 0 | −0.0923 | 0.2714 |
| Ang-2 | −0.0246 | 0.1193 | | | 21.8 | 1.0047 | 62.8642 | 0 | −0.2792 | 0.3008 |

*As IP-10 is kept as non-linear effect using natural cubic splines with three knots, there are two terms of this variable in the model.

RMSD: root mean squared difference.

## Appendix 8

### Results when including viremia as a potential biomarker

We included plasma viremia (plasma viral RNA) levels along with the ten biomarkers and performed analyses similar to the main analyses.

The single and global models showed that higher viremia was associated with increased risk of severe or moderate dengue (*Appendix 8—figure 1*, *Appendix 8—table 1*). The association was relatively linear with log-10 viremia levels and was not different between children and adults. This finding is similar to our previous study which shows that higher plasma viremia levels are associated with increased more severe outcome in dengue infection, regardless of age, serotype and host immune status (*Vuong et al., 2021a*). Results of the association between viremia and the endpoint were similar between the single and global models. In addition, including viremia to the global model did not affect markedly to the results of other biomarkers. This suggests that viremia and the other ten biomarkers might not be confounders or intermediate variables of each other in the association with severe or moderate dengue outcome.

Results from the 'best subset' procedure when including viremia as a potential biomarker showed that viremia was not selected in any of the best combinations in children. The results for children were the same with the main analysis (*Appendix 8—table 2*). For adults, the selection resulted differently: the best of all combinations included five biomarkers SDC-1, IL-8, ferritin, viremia and sCD163. The best combinations of 2 and 3 variables were the same with the main analysis. Viremia was selected in the best combination of 4 and 5 biomarkers (*Appendix 8—table 3*).

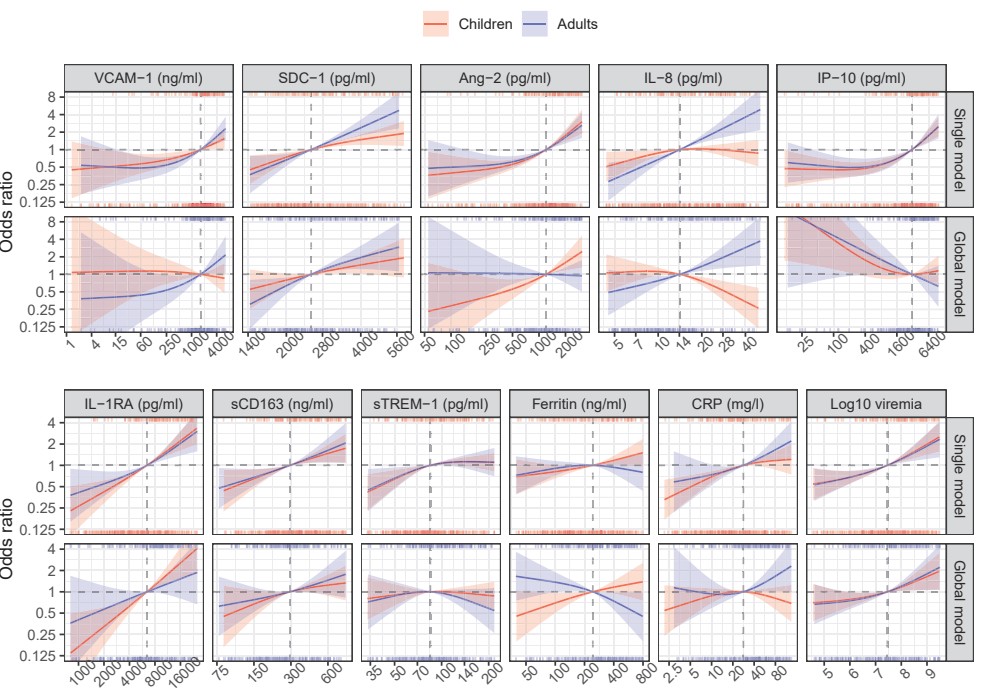

**Appendix 8—figure 1.** Results from models for severe/moderate dengue including viremia as a potential biomarker.

**Appendix 8—table 1.** Results from models for severe/moderate dengue including viremia as a potential biomarker.

| | Single models | | | | Global model | | | |
| --- | --- | --- | --- | --- | --- | --- | --- | --- |
| | Children OR (95% CI) | Adults OR (95% CI) | $P_{overall}$ | $P_{interaction}$ | Children OR (95% CI) | Adults OR (95% CI) | $P_{overall}$ | $P_{interaction}$ |
| VCAM-1 (ng/ml) | | | <0.001 | 0.715 | | | 0.286 | 0.136 |
| - 1636 vs 818 | 1.20 (1.04–1.38) | 1.35 (1.15–1.58) | | | 0.94 (0.76–1.17) | 1.34 (1.03–1.74) | | |
| - 3272 vs 1636 | 1.25 (1.02–1.53) | 1.48 (1.19–1.85) | | | 0.93 (0.69–1.24) | 1.45 (1.02–2.04) | | |
| SDC-1 (pg/ml) | | | <0.001 | 0.088 | | | 0.005 | 0.645 |
| - 2519 vs 1260 | 2.67 (1.31–5.43) | 3.33 (1.32–8.42) | | | 2.07 (0.78–5.47) | 4.28 (1.27–14.43) | | |
| - 5039 vs 2519 | 1.71 (1.18–2.47) | 3.71 (2.09–6.58) | | | 1.71 (0.95–3.09) | 2.55 (1.17–5.57) | | |
| Ang-2 (pg/ml) | | | <0.001 | 0.524 | | | 0.060 | 0.070 |
| - 1204 vs 602 | 1.64 (1.39–1.94) | 1.51 (1.26–1.82) | | | 1.62 (1.19–2.20) | 0.97 (0.71–1.34) | | |
| - 2409 vs 1204 | 2.21 (1.58–3.10) | 2.00 (1.40–2.85) | | | 1.92 (1.22–3.01) | 0.96 (0.61–1.49) | | |
| IL-8 (pg/ml) | | | <0.001 | <0.001 | | | <0.001 | <0.001 |
| - 14 vs 7 | 1.42 (1.05–1.91) | 2.18 (1.47–3.24) | | | 0.89 (0.61–1.31) | 1.60 (0.99–2.59) | | |
| - 28 vs 14 | 0.99 (0.78–1.25) | 2.33 (1.63–3.33) | | | 0.52 (0.36–0.77) | 1.96 (1.28–3.02) | | |
| IP-10 (pg/ml) | | | <0.001 | 0.984 | | | 0.150 | 0.500 |
| - 3093 vs 1546 | 1.46 (1.26–1.68) | 1.45 (1.21–1.73) | | | 0.93 (0.73–1.19) | 0.75 (0.53–1.06) | | |
| - 6186 vs 3093 | 1.68 (1.35–2.09) | 1.69 (1.29–2.22) | | | 1.07 (0.76–1.50) | 0.75 (0.47–1.20) | | |
| IL-1RA (pg/ml) | | | <0.001 | 0.082 | | | <0.001 | 0.062 |
| - 6434 vs 3217 | 1.69 (1.42–2.03) | 1.48 (1.21–1.81) | | | 1.97 (1.42–2.73) | 1.40 (0.93–2.09) | | |
| - 12868 vs 6434 | 1.82 (1.46–2.27) | 1.70 (1.29–2.24) | | | 2.03 (1.41–2.92) | 1.38 (0.87–2.19) | | |
| sCD163 (ng/ml) | | | <0.001 | 0.551 | | | 0.124 | 0.289 |
| - 295 vs 147 | 1.57 (1.14–2.15) | 1.49 (1.13–1.98) | | | 1.51 (0.94–2.42) | 1.30 (0.86–1.98) | | |
| - 589 vs 295 | 1.46 (1.10–1.93) | 1.61 (1.09–2.37) | | | 1.24 (0.88–1.73) | 1.44 (0.91–2.28) | | |
| sTREM-1 (pg/ml) | | | 0.059 | 0.997 | | | 0.745 | 0.594 |
| - 85 vs 42 | 1.87 (1.23–2.84) | 1.79 (1.10–2.93) | | | 1.16 (0.71–1.91) | 1.24 (0.65–2.36) | | |
| - 169 vs 85 | 1.12 (0.91–1.38) | 1.12 (0.82–1.53) | | | 0.92 (0.67–1.27) | 0.67 (0.41–1.10) | | |
| Ferritin (ng/ml) | | | 0.042 | 0.054 | | | 0.007 | 0.002 |
| - 243 vs 122 | 1.18 (1.01–1.38) | 1.06 (0.89–1.27) | | | 1.32 (1.04–1.67) | 0.77 (0.60–0.98) | | |
| - 487 vs 243 | 1.26 (1.00–1.58) | 0.90 (0.66–1.23) | | | 1.22 (0.89–1.68) | 0.64 (0.42–0.98) | | |
| CRP (mg/l) | | | <0.001 | 0.031 | | | 0.185 | 0.113 |
| - 28 vs 14 | 1.26 (1.12–1.41) | 1.25 (1.03–1.52) | | | 1.06 (0.91–1.23) | 1.09 (0.84–1.43) | | |
| - 56 vs 28 | 1.13 (0.95–1.34) | 1.38 (1.11–1.71) | | | 0.89 (0.72–1.11) | 1.35 (1.01–1.81) | | |
| Log-10 viremia (copies/ml) | | | <0.001 | 0.747 | | | 0.040 | 0.886 |
| - 7.5 vs 6.5 | 1.34 (1.18–1.53) | 1.33 (1.16–1.53) | | | 1.21 (1.03–1.42) | 1.25 (1.04–1.51) | | |
| - 8.5 vs 7.5 | 1.53 (1.25–1.87) | 1.48 (1.16–1.88) | | | 1.35 (1.05–1.74) | 1.43 (1.05–1.95) | | |

Odds ratios are estimated at age of 10 and 25 years, represented as children and adults respectively; $P_{overall}$ is derived from Wald test for the overall association of the biomarker with the endpoint; $P_{interaction}$ is from the test for the interaction between the biomarker and age.

**Appendix 8—table 2.** Best combinations of biomarkers associated with severe or moderate dengue for children.

| | Best of all combinations | Best combination of 2 variables | Best combination of 3 variables | Best combination of 4 variables | Best combination of 5 variables |
|---|---|---|---|---|---|
| Variables | | | | | |
| - VCAM-1 | | | | | |
| - SDC-1 | + | | | | |
| - Ang-2 | + | | + | + | + |
| - IL-8 | + | | | | + |
| - IP-10 | + | | | + | + |
| - IL-1RA | + | + | + | + | + |
| - sCD163 | | | | | |
| - sTREM-1 | | | | | |
| - Ferritin | + | + | + | + | + |
| - CRP | | | | | |
| - Viremia | | | | | |
| AIC of the selected model | 465.9 | 484.7 | 480.0 | 473.7 | 467.6 |

VCAM-1: vascular cell adhesion molecule-1; SDC-1: syndecan-1; Ang-2: angiopoietin-2; IL-8: interleukin-8; IP-10: interferon gamma-induced protein-10; IL-1RA: interleukin-1 receptor antagonist; sCD163: soluble cluster of differentiation 163; sTREM-1: soluble triggering receptor expressed on myeloid cells-1; CRP: C-reactive protein; AIC: Akaike information criterion.

**Appendix 8—table 3.** Best combinations of biomarkers associated with severe or moderate dengue for adults.

| | Best of all combinations | Best combination of 2 variables | Best combination of 3 variables | Best combination of 4 variables | Best combination of 5 variables |
|---|---|---|---|---|---|
| Variables | | | | | |
| - VCAM-1 | | | | | |
| - SDC-1 | + | + | + | + | + |
| - Ang-2 | | | | | |
| - IL-8 | + | + | + | + | + |
| - IP-10* | | | | | |
| - IL-1RA | | | | | |
| - sCD163 | + | | | | + |
| - sTREM-1 | | | | | |
| - Ferritin | + | | + | + | + |
| - CRP | | | | | |
| - Viremia | + | | | + | + |
| AIC of the selected model | 426.4 | 441.1 | 434.2 | 428.5 | 426.4 |

VCAM-1: vascular cell adhesion molecule-1; SDC-1: syndecan-1; Ang-2: angiopoietin-2; IL-8: interleukin-8; IP-10: interferon gamma-induced protein-10; IL-1RA: interleukin-1 receptor antagonist; sCD163: soluble cluster of differentiation 163; sTREM-1: soluble triggering receptor expressed on myeloid cells-1; CRP: C-reactive protein; AIC: Akaike information criterion.

*Variable is kept as non-linear effect using natural cubic splines with three knots.

## Appendix 9

## Results of variable selection using different approaches

**Appendix 9—table 1.** Results of variable selection for children.

|  | VCAM-1 | SDC-1 | Ang-2 | IL-8 | IP-10 | IL-1RA | sCD163 | sTREM-1 | Ferritin | CRP |
|---|---|---|---|---|---|---|---|---|---|---|
| Best subset |  | + | + | + | + | + |  |  | + |  |
| Backward elimination |  | + | + | + | + | + |  |  | + |  |
| Forward selection |  | + | + | + | + | + |  |  | + |  |
| Stepwise forward |  | + | + | + | + | + |  |  | + |  |
| Stepwise backward |  | + | + | + | + | + |  |  | + |  |
| Augmented backward elimination | + | + | + | + | + | + |  |  | + |  |
| Bayesian projection |  |  | + | + | + | + |  |  | + |  |

**Appendix 9—table 2.** Results of variable selection for adults.

|  | VCAM-1 | SDC-1 | Ang-2 | IL-8 | IP-10* | IL-1RA | sCD163 | sTREM-1 | Ferritin | CRP |
|---|---|---|---|---|---|---|---|---|---|---|
| Best subset |  | + |  | + | + | + | + | + | + |  |
| Backward elimination |  | + |  | + | + | + | + | + | + |  |
| Forward selection |  | + |  | + |  | + |  | + | + |  |
| Stepwise forward |  | + |  | + |  | + |  | + | + |  |
| Stepwise backward |  | + |  | + | + | + | + | + | + |  |
| Augmented backward elimination | + | + |  | + | + | + | + | + | + |  |
| Bayesian projection |  | + |  | + |  |  |  |  |  |  |

*Variable is kept as non-linear effect using natural cubic splines with three knots.

