## [Decision Letter]

**Acceptance summary:**

The paper is relevant as dengue diagnosis and prognosis remains a major problem in several parts of the world. This paper should help the development of biomarker panels for clinical use and could improve triage and risk prediction in dengue patients.

**Decision letter after peer review:**

Thank you for submitting your article "Combination of inflammatory and vascular markers in the febrile phase of dengue is associated with more severe outcomes" for consideration by *eLife*. Your article has been reviewed by 3 peer reviewers, and the evaluation has been overseen by Balram Bhargava as the Senior and Reviewing Editor. The reviewers have opted to remain anonymous.

Essential revisions:

Please address the weaknesses of the study which have been highlighted in the reviewers comments as below.

*Reviewer #2 (Recommendations for the authors):*

Page 8, line 5. Could the authors provide a rationale on why a 1 case:2 control ratio was chosen for this study?

Table 1. The inclusion of results from statistical tests would be useful to accompany the description provided in page 12, lines 1-6.

Table 1. Is the biomarker validation affected by the preponderance of DENV-1 in the study population? Although the number of dengue cases caused by the other serotypes is relatively lower, is it possible to show that at least in terms of trends, there is no difference in the expression of the biomarkers by serotype? This could be a very useful supplementary information to address any concern on whether the findings reported here could be applied to dengue cases caused by non-DENV-1 infection.

Table 1. Most of the severe dengue cases were in children whereas there were proportionately more moderately severe cases in adults. Could the different sets of biomarkers for paediatric and adult dengue cases be predictive for severe vs moderately severe dengue?

Tables 2 and 3 can be shown as supplementary tables since the same datasets are shown as Figures 2 and 3. It may be busy but it would also be useful to indicate in Figure 2 where the statistically significant differences are.

Page 17, line 8. I do not understand how the global model was developed and how it was useful for biomarker development. Indeed, the example highlighted in this paragraph was IP-10, where the direction of correlation changed in the global compared to single model. Despite this finding, Tables 4 and 5 went on to consider IP-10 as amongst the possible biomarkers to predict severe/moderately severe dengue. Please provide a clearer explanation for readers without deep expertise in statistics to appreciate the findings reported here.

The lack of viraemia and NS1 antigenaemia measurements is a striking omission. I think it would be of great advantage to include these parameters into the analysis. If not possible, the authors should provide a discussion on why these parameters were excluded and whether they should be considered for future studies.

*Reviewer #3 (Recommendations for the authors):*

1. The model can be confusing to understand. it appears to this non statistician reviewer that global model did not improve sensitivity of the test? If so, how does that impact the interpretation of the data?

2. Certain sections of the discussion are repetitive and can be more succinctly summarised.

---

## [Author Response]

Reviewer #2 (Recommendations for the authors):Page 8, line 5. Could the authors provide a rationale on why a 1 case:2 control ratio was chosen for this study?

As there were a limited number of cases with the primary endpoint (severe/moderate dengue) in the clinical study, our aim was to increase the power of the study by doubling the number of controls (uncomplicated dengue).

Table 1. The inclusion of results from statistical tests would be useful to accompany the description provided in page 12, lines 1-6.

We did not perform statistical tests for table 1 because of the following:

– This was for descriptive analysis only and there was no hypothesis to test

– Performing statistical analysis for table 1 would mean that the independent variable is severity (severe/moderate versus uncomplicated dengue) and the outcomes are the clinical characteristics, which reverses the causal pathway.

Table 1. Is the biomarker validation affected by the preponderance of DENV-1 in the study population? Although the number of dengue cases caused by the other serotypes is relatively lower, is it possible to show that at least in terms of trends, there is no difference in the expression of the biomarkers by serotype? This could be a very useful supplementary information to address any concern on whether the findings reported here could be applied to dengue cases caused by non-DENV-1 infection.

Thank you – please see our response to the public review for reviewer 2, the results of the single and global models are similar between DENV-1 and non-DENV-1 infections.

Table 1. Most of the severe dengue cases were in children whereas there were proportionately more moderately severe cases in adults. Could the different sets of biomarkers for paediatric and adult dengue cases be predictive for severe vs moderately severe dengue?

Because the number of severe dengue cases is limited (38 cases), it is not possible to analyse severe versus moderate dengue. In our analysis for severe dengue alone (severe versus moderate/uncomplicated dengue) as shown in appendix 6 (figure S5 and table S6) in the supplementary file, we could see a trend in some of the biomarkers, but confidence intervals were very wide.

Tables 2 and 3 can be shown as supplementary tables since the same datasets are shown as Figures 2 and 3. It may be busy but it would also be useful to indicate in Figure 2 where the statistically significant differences are.

We agree that table 2 is similar to figure 2 and we have moved it to the supplementary file (table S3). However, we would like to keep table 3 in the main file (which is changed to table 2 of the revised manuscript) as it contains important and distinct results which are not represented in figure 3, such as numerical results, p-values for the overall effect and the interaction between biomarkers and age.

Similar to our second response to reviewer 2 above, we did not perform statistical tests for figure 2 as this is for descriptive analysis only; we do not want to test for differences in trend over time. Also, performing tests would result in reversal of the causal pathway, with the biomarker as dependent variable and dengue outcome as predictor. Note that the relationships with p-values and confidence intervals can be found in table 2 and figure 3 of the revised manuscript (P values).

Page 17, line 8. I do not understand how the global model was developed and how it was useful for biomarker development. Indeed, the example highlighted in this paragraph was IP-10, where the direction of correlation changed in the global compared to single model. Despite this finding, Tables 4 and 5 went on to consider IP-10 as amongst the possible biomarkers to predict severe/moderately severe dengue. Please provide a clearer explanation for readers without deep expertise in statistics to appreciate the findings reported here.

When performing variable selection, a backward approach is preferable over a forward approach. Therefore, we started variable selection with a global model that contained all the biomarkers along with the interaction with age. The global model allows investigation of the association between biomarkers and the clinical endpoint when considering them together (aim #1). It was the initial step to develop a set of biomarkers that is most strongly associated with the primary endpoint (aim #2). We have added this point to the Statistical analysis section of the revised manuscript (page 9, line 25 and page 10, lines 1-2).

We fitted the single models because it may provide further information on disease mechanisms when compared with results from the global model (although not our primary study aim). The observation that the direction of the association between IP-10 and the endpoint changed in the global model compared to single model suggests that there might be confounders or intermediate variables of IP-10 in the global model. It could be because of IL-1RA since both markers show a fairly strong correlation (Spearman’s correlation coefficient was 0.75). However, changing the direction of the association from the single to global model does not diminish the possibility of that biomarker being selected in the best combinations. See also the reply to comment #3 of reviewer #1. This explanation was added to the Discussion section of the revised manuscript (page 14, lines 10-17).

The lack of viraemia and NS1 antigenaemia measurements is a striking omission. I think it would be of great advantage to include these parameters into the analysis. If not possible, the authors should provide a discussion on why these parameters were excluded and whether they should be considered for future studies.

Thank you – please see our response to the public review for reviewer 2. We have added viremia data, but NS1 antigenaemia data was not available.

Reviewer #3 (Recommendations for the authors):1. The model can be confusing to understand. it appears to this non statistician reviewer that global model did not improve sensitivity of the test? If so, how does that impact the interpretation of the data?

Thank you, please also see our response to comment #10 of reviewer #2. The global model is the model to start with and contains all the biomarkers combined, taking into account the interaction with age. In this study, building the global model is important because: (1) it allows investigation into the association between biomarkers and clinical endpoint when considering them together (aim #1), and (2) it was the first step to develop a set of biomarkers that is most strongly associated with the primary endpoint (aim #2). We did not analyse the predictive performance of the models. Choosing a cutoff value for the score to compute the sensitivity is better done in a separate study (see also response to comment #4 of reviewer #1).

2. Certain sections of the discussion are repetitive and can be more succinctly summarised.

Thank you for this suggestion, we have revised and cut down the discussion to make it more concise.